## Special feature review

evolution, genetics

epigenetic variation, DNA methylation, histone modifications, genetic variation, natural populations, phenotypic variation

**Author for correspondence:**
Arild Husby
e-mail: arild.husby@ebc.uu.se

One contribution to a special feature 'Wild Quantitative Genomics: the genomic basis of fitness variation in natural populations'. Guest edited by Susan Johnston, Nancy Chen, Emily Josephs.

# Wild epigenetics: insights from epigenetic studies on natural populations

Arild Husby[1,2]

[1]Evolutionary Biology, Department of Ecology and Genetics, Uppsala University, Norbyvägen 18D, SE-75236 Uppsala, Sweden
[2]Centre for Biodiversity Dynamics, Norwegian University for Science and Technology, Trondheim, Norway

 AH, 0000-0003-1911-8351

Epigenetic mechanisms such as DNA methylation, histone modifications and non-coding RNAs are increasingly targeted in studies of natural populations. Here, I review some of the insights gained from this research, examine some of the methods currently in use and discuss some of the challenges that researchers working on natural populations are likely to face when probing epigenetic mechanisms. While studies supporting the involvement of epigenetic mechanisms in generating phenotypic variation in natural populations are amassing, many of these studies are currently correlative in nature. Thus, while empirical data point to widespread contributions of epigenetic mechanisms in generating phenotypic variation, there are still concerns as to whether epigenetic variation is instead ultimately controlled by genetic variation. Disentangling these two sources of variation will be a key to resolving the debate about the importance of epigenetic mechanisms, and studies on natural populations that partition the relative contribution of genetic and epigenetic factors to phenotypic variation can play an important role in this debate.

## 1. Technology advances epigenetic studies in natural populations

There has been a rapid increase in the number of studies examining epigenetic factors in wild animal populations over the last few years (e.g. [1,2–7]). In this review, I outline some of the reasons for this trend, summarize the insights gained from this research, look at how we can measure epigenetic mechanisms, and point out some caveats, but also opportunities, that come with the increased ease of obtaining epigenetic information on wild populations.

Epigenetic mechanisms (DNA methylation, histone modification, small RNAs) are now increasingly being targeted in studies of natural populations, no doubt at least in part as a result of the ease with which technological advances allow probing of epigenetic modulations [8]. Pioneering work in the 1960s established the central role of histone acetylation for regulation of gene expression [9], and by the mid-2000s, the first large-scale mapping atlases of epigenomes were published [10] by combining chromatin immunoprecipitation with microarrays. Subsequent technological advancements (ChIP-seq (chromatin immunoprecipitation) and ATAC-seq (assay for transposase-accessible chromatin)) mean that it is now possible to obtain information about chromatin structure in non-model species at a genome-wide scale [11,12]. A similar technological development has driven the increase in profiling individuals for DNA methylation variation following the discovery that methylation-sensitive restriction enzymes could be applied to examine DNA methylation patterns in the 1970s [13–15]. The combination of restriction enzymes with high-throughput sequencing now allows for nucleotide-level resolution of DNA methylation information from tens to hundreds of individuals in natural

populations and has become the most widely used technique to obtain DNA methylation variation from natural populations [4,5,16].

Parallel with these technological advancements, there has also been a growing recognition and increased awareness that epigenetic mechanisms can play a key role in generating phenotypic diversity [17–19] and that for a full understanding of the significance of epigenetic mechanisms, the ecological context in which they are studied is important [17]. So, what have we learned from ecological epigenetic studies, and what are the main unanswered questions and challenges that remain?

## 2. Insights from epigenetic studies on wild populations

One of the first demonstrations that epigenetic mechanisms can play a key role in generating phenotypic diversity in a natural population came from the plant common toadflax (*Linaria vulgaris*) [20]. In fact, this plant has a long history of upending established views: Carl Linnaeus discovered in the 1740s a flower, unknown to him, that looked identical to common toadflax but that displayed radial symmetry in the flower arrangement instead of the dorsoventral asymmetry normally seen. This caused Linnaeus much trouble as it challenged the view of species as immutable entities that were common at the time, and he named this plant Peloria, which is Greek for 'monster', and wrote that this finding was 'no less surprising than if a cow had given birth to a calf with a wolfs head' [21]. Two hundred and fifty years later, Cubas *et al.* [20] demonstrated that this polymorphism is the result of differences in DNA methylation levels at the *cycl* gene and thus that the peloric (unusual symmetry) flowers are, at least in part, under epigenetic control. This finding highlighted that epigenetic factors can be important mechanisms for generating phenotypic diversity in natural populations.

Following this finding, there are now many demonstrations that epigenetic mechanisms are involved in producing phenotypes as diverse as eye development in cavefish [22], body size variation in carpenter ants [23], salinity adaptation in sticklebacks [4] and seasonal timing of reproduction in great tits [24]. This is not to say that genetic variation is unimportant in these examples (see [25] for a recent critique), but epigenetic mechanisms can clearly contribute to the generation of adaptive phenotypic variation.

Some of the insights emerging from ecological epigenetic studies are also shared among different species and thus point to a conserved epigenetic mechanism, as in the case of the role of DNA methylation and histone modifications in controlling flowering time in several species of plants for example. The transition from the vegetative to flowering state occurs in response to environmental cues such as day length, but in many species exposure to cold temperatures (vernalization) is also needed. Use of demethylating agents on *Arabidopsis thaliana* and *Thlaspi arvense* first demonstrated that low temperature treatment results in demethylation but also that demethylation leads to induction of flowering [26]. DNA methylation therefore provides a mechanism by which expression of key genes, such as the flowering locus C (*FLC*), is inhibited to prevent early flowering. Later research demonstrated that the methylation level changes at *FLC* are due to histone modifications, and these contribute

to natural variation in the vernalization response in different *Arabidopsis* accessions [27]. This mechanism has since been demonstrated also in other related plant species [28].

Environmental influences on epigenetic mechanisms as in the example above have attracted particular interest as a potential explanation for how phenotypic plasticity is generated [29]. One example where the epigenetic mechanism behind an environmentally induced phenotypic trait has been resolved is the plastic response in sex determination in relation to temperature in turtles. Ge *et al.* [30] found that knockdown of a histone demethylase gene (a protein that removes methyl groups) that was consistently differentially expressed across development at different temperatures promotes transcription of the male sex determining gene *Dmrt1* through demethylation of another histone (H3K27me3) in the promotor region of *Dmrt1*. This is one of few studies on natural populations that have used functional assays to demonstrate the involvement of an epigenetic mechanism in an environmentally induced trait.

Whether epigenetic modifications are transmitted across generations independent of sequence variants is a hotly debated topic [31–34], and many technical challenges remain in order to fully rule out potential confounding effects [25]. As both experimental design and technical issues are remedied in future studies, we should emerge with a clearer idea of the potential generality of transgenerational epigenetic effects, and here, studies on natural populations can play a key role (see below).

## 3. How to measure epigenetic mechanisms in natural populations

There are several techniques available to those interested in measuring epigenetic mechanisms, and which ones to use depend on the goals of the study as well as the available budget. Below, I briefly review the most used technological approaches to obtain epigenetic data from natural populations, focusing mainly on techniques for measuring DNA methylation.

### (a) Obtaining DNA methylation information from reduced representation approaches

Bisulfite conversion of libraries prior to sequencing allows inferences to be made about DNA methylation because bisulfite converts unmethylated cytosine to uracil, unlike methylated cytosines which are resistant to this deamination. This thereby allows for direct reading of DNA methylation state genome-wide at nucleotide-level resolution [35]. I will first discuss the use of reduced representation approaches, which is the most frequently used technique in natural populations [16].

Reduced representation bisulfite sequencing (RRBS) is a method by which genomic DNA is digested with methylation-insensitive restriction enzymes (normally MspI but TaqI can also be used) before end repair, polyA-tailing and adapter ligation. This is then followed by size selection (see §4*a* for more on this) and bisulfite conversion before sequencing the libraries [36]. The use of MspI (CCGG) or TaqI (TCGA) ensures the presence of CpG sites in every read, and the expected numbers, sizes and sequences of fragments can be predicted by in silico digestion of the genome of the

species in question or in some cases also a closely related species with a reference genome. The RRBS approach is similar to that used in whole-genome bisulfite sequencing except the use of restriction enzymes means RRBS is enriched for CpG-dense regions of the genome such as CpG islands and promoter regions [35,36]. Sequence reads are then mapped to a bisulfite-converted reference genome using specific mapping software that considers the effect of the bisulfite conversion. RRBS strongly benefits from a reference genome, which can obviously restrict its use for studies of natural populations for which there may be none in many cases, because functional interpretation is otherwise difficult.

Another reduced representation approach to obtain methylation information on individual CpG sites is epiGBS (epigenetic genotyping by sequencing) [37]. Like RRBS, this method also deploys restriction enzymes (PstI) to digest the genome, but this method does not require a reference genome as it does a de novo reference construction and mapping to call DNA methylation levels [37]. Apart from the obvious benefit of not needing a reference genome, an additional advantage of the epiGBS approach is that it is not enriched for CpG rich regions of the genome, and thus this method can target a wider range of organisms for which other types than CpG methylation is important, such as in plants [38].

Yet another reduced representation approach is epigenetic restriction site associated DNA (RAD) sequencing [39], which uses a combination of restriction enzymes (it is a modified double digest RAD protocol) where one restriction enzyme is the methylation-sensitive HpaII. This method samples only loci that do not contain 5 mC bases, and it is possible to infer methylation state by comparing the frequencies at which loci are sampled between groups [39]. This method can be used both with and without a reference genome.

All reduced representation approaches of course trade off costs versus genomic coverage, but scalability of projects is achieved easier compared to whole-genome sequencing, and thus they allow sampling of many more individuals. This is important given concerns regarding statistical power in ecological (epi)genetic studies [40].

## (b) DNA methylation information from whole-genome sequencing approaches

Information about DNA methylation at the level of whole genomes can be obtained through several different methodologies of which whole-genome bisulfite sequencing (WGBS) and long read sequencing technology (PacBio and Oxford nanopore) are most common. Since there is no use of restriction enzymes in these approaches, this also offers the possibility to test for non-CpG methylation as has been done in the ecological model species the great tit (Parus major) [41] for example. Other examples of the use of WGBS on natural populations include work on three-spined sticklebacks by Metzger & Schulte [42] to examine the DNA methylation landscape in relation to evolutionary strata on the sex chromosomes and the role methylation may have in adaptation to salinity variation by Heckwolf et al. [4]. In Mexican cavefish, WGBS was used to pinpoint that DNA methylation variation at the promoter in several eye development genes is responsible for eye degeneration [22].

While bisulfite treatment offers a convenient way to obtain information about methylation variation at individual

nucleotide sites across the genome, bisulfite treatment is a very harsh chemical treatment of the DNA that can lead to substantial fragmentation [43]. Fragmentation can be a particular issue for low-input DNA such as that from museum samples [44], and alternative (bisulfite-free) methods have been developed [45], but not yet used on wild populations to my knowledge.

Long read sequencing methods such as the single molecule real time (SMRT) approach by Pacific Biosciences and the nanopore sequencing method by Oxford Nanopore Technologies (ONT) can directly obtain information about methylation status at individual nucleotide sites without chemical treatment. In SMRT sequencing, base modifications are inferred by the delay between fluorescence pulses, whereas in ONT, base modifications are detected by the difference in flow of the current through the pore which results in signal shifts [46]. At present, these approaches need high coverage and suffer from high false discovery rates [46], but given the advancement in both the software for calling base modifications as well as in the sequencing technologies themselves, these methods may well be a feasible approach for larger-scale assays soon. For example, the introduction of the HIFI approach by PacBio [47] where each molecule is sequenced several times has dramatically improved error rates to the levels observed with short read technology and will probably contribute to improved calling of base modifications as well.

A summary of some of the advantages and disadvantages for different methods used to study DNA methylation pattern can be found in table 1.

## (c) Measuring chromatin and histone modifications

It has long been known that acetylation and methylation of histones and thus the dynamics of chromatin modifications play an important part in gene regulation [9]. Early assays to examine chromatin landscapes involved tedious and complicated cell preparations and required very high input material, but development of ATAC-seq [48] can now be used also on non-model species where less input material is often available. Indeed, this method has already been used on natural populations of Heliconius butterflies to examine the chromatin landscape, focusing particularly around the optix gene that controls wing pattern mimicry [12].

Molecular studies on sex determination in turtles have shown that histone modifications can be important in determining phenotypic variation in traits that are relevant for population demographic parameters. In many reptile species, sex is determined by ambient temperature during development. Ge et al. [30] followed up earlier reports of correlations between DNA methylation and histone modification changes in relation to environmentally induced sex determination in the red-eared slider turtle (Trachemys scripta elegans) using a functional approach (RNA interference). The Kdm6b gene is histone demethylase (a protein that removes methyl groups), and knockdown of this gene promotes the transcription of the male sex-determining gene Dmrt1 through demethylation of a histone (H3K27me3) in the promotor region of Dmrt1. The study by Ge et al. [30] therefore convincingly demonstrates at the functional level how environmentally induced variation in a trait (sex determination) is determined by an epigenetic mechanism, in this case through histone modification.

**Table 1.** Overview of the advantages and disadvantages for some common methods to measure DNA methylation.

| method | advantages | disadvantages |
|---|---|---|
| EPI-GBS | inexpensive, can target also non-CpG methylation | needs methylated adapters, in absence of reference genome need to sequence library prior to bisulfite treatment to build the reference |
| EPIRADSEQ | does not require a reference genome, easy to scale in terms of nr of loci | needs methylated adapters, only gives information about methylation state of the *HpaII* cut site, genetic variation in restriction sites between individuals can lead to allele specific dropout. Not widely adopted by the community |
| reduced representation bisulfite sequencing (RRBS) | low input requirement, single nucleotide resolution, very cost efficient relative to WGBS, methylation information at different sequence contexts (CG, CHG, CHH) | needs methylated adapters, sparse sampling of the genome, strongly benefits from a reference genome, DNA fragmentation and reduced sequence complexity due to bisulfite conversion, SNPs where a cytosine is converted to thymidine will be masked after bisulfite conversion, difficult to infer DMRs (differentially methylated regions) |
| Whole-genome bilsuphite sequencing (WGBS) | low input requirement, single nucleotide resolution, methylation information at different sequence contexts (CG, CHG, CHH) | needs methylated adapters, needs a reference genome, expensive (especially for large genomes), DNA fragmentation and reduced sequence complexity due to bisulfite conversion, SNPs where a cytosine is converted to thymidine will be masked after bisulfite conversion |

## 4. Some technical considerations for epigenetic studies on natural populations

### (a) Fragment size selection and impact on alignment success in reduced-representation bisulfite sequencing data

Given the popularity of RRBS method in natural populations [16], I will here examine one important design consideration that is sometimes overlooked: how fragment size selection impacts subsequent alignment success. In mammals, fragment sizes of 40–220 bp have been used [49], and in birds in the range of 20–200 bp [5], but why does it matter which fragment size selection is applied? As we saw above, MspI cuts 'CCGG' motifs which generates variation in the fragment sizes between cut sites which are specific to the organism being studied. Thus, with access to a reference genome, one can do *in silico* digestion with MspI to examine the fragment size distribution and look for fragment sizes that are overrepresented, and thus probably map to repetitive regions of the genome. Considering the cost of sequencing and the desire to have a certain coverage (see below), including many fragments from repetitive regions in the library will lead to lower mapping efficiency since they will not map at all or will not map uniquely. This again will lead to lower coverage and fewer CpG sites for the statistical analysis. I am not aware of comparisons where *in silico* digestions of the genome with restriction enzymes from a wild organism have been published, but it would be very useful to have information about this (e.g. in the electronic supplementary material), so others can understand why (or why not) a

particular fragment size range was chosen. An interesting study on pigs has demonstrated the problems that sequencing libraries with fragment sizes containing repetitive elements can lead to for subsequent mapping and coverage [49].

In addition to the problem of having potential repetitive elements in the fragment size selection, one should also make sure that the fragments sequenced are not too short as this may also negatively impact mapping success given that after bisulfite conversion, one is left with only three bases for use in alignment.

In conclusion, if you have access to a reference genome of the organism of interest or a closely related species, then *in silico* digestion with MspI should be used to optimize size selection in the RRBS protocol and to avoid potential repetitive elements to make sure sufficient coverage of CpG sites is available for downstream analyses.

### (b) Coverage recommendations

Sequencing depth and coverage are of course of key concern in any genomic analysis [50], and this applies also when doing DNA methylation analyses. Ziller *et al.* [51] used available WGBS datasets to derive minimum sequencing depth suggestions for this method and found that sequencing each sample at higher than 10× did not lead to much higher sensitivity and that instead increasing the number of replicates has correspondingly more improvement on power [51]. This finding has largely been confirmed in a simulation study [40] which found that above 15–20× coverage, the increase in power is negligible. This is in contrast to the benefit of adding more samples, and indeed many

ecological epigenetic studies are underpowered in terms of the number of individuals included for detecting differentially methylated sites or regions [40].

## (c) Storage of samples and the impact on methylation levels

For most studies of natural populations, DNA is not extracted from the relevant tissue at the time of sampling, but instead samples are used that have been stored for various amounts of time in different ways and solutions. Storage method and time are known to influence DNA/RNA yield, and one obvious question of concern is if storage time and method can impact methylation levels, and if so, in what direction. This is particularly important for methylation studies since DNA methylation can be modulated by different environmental factors.

Schroder & Steimer [52] examined the impact of storing blood from humans at different temperatures and length of time on DNA yield and DNA methylation patterns of one gene (*HIF3A*). Blood samples were either used for DNA extraction and methylation sequencing immediately, or blood was stored at −70°C, −20°C, 2–8°C or room temperature and then DNA was extracted after one, three and ten months. Additionally, one blood sample was frozen at −70°C but then thawed and re-frozen on a weekly basis. For all storage conditions, the DNA yield after extraction decreased with storage time. By contrast, overall DNA methylation increased with storage time by between 2% and 9% after 10 months [52]. Interestingly, it was the samples stored in a refrigerator (2–8°C) that showed the least change in DNA methylation levels. However, when the authors examined changes in methylation at individual CpG sites instead of overall DNA methylation levels, there were very large differences among sites in the temporal change depending on the methylation level at the onset. Sites with low DNA methylation levels (less than 20%) had a large increase in methylation with storage time (43% increase on average), whereas sites with high DNA methylation levels (greater than 20%) showed less of an increase (29% on average). The blood samples were stored with an anticoagulant (EDTA), and so whether these results are general or specific to the storage method is not clear; another study also found increases in DNA methylation level (in white blood cells) with storage time over 15 days [53]. By contrast, a recent review paper [54] examining many different storage methods concluded that there are negligible effects of storage method on DNA yield and overall DNA methylation levels, although they also cautioned that there can be CpG site-specific changes to DNA methylation levels.

Taken together, there is a clear concern that differences in storage time and method among samples may induce 'artificial' changes in DNA methylation patterns at individual CpG sites, thereby making it more difficult to reach general conclusions on the potential role that changes in DNA methylation may have on phenotypes. Careful description of storage time and method in epigenetic studies on natural populations is therefore needed, and when using samples stored for varying amounts of time (or solutions), one should strive to include a sufficient number of technical replicates to control for such sample storage variation.

## (d) Tissue type used in DNA methylation studies on natural animal populations

Due to the ease of sampling, blood is the most commonly used tissue type when examining epigenetic variation [16]. However, there are many important considerations other than convenience to keep in mind when choosing which tissue to profile for epigenetic variation. Unlike genetic variation, which is identical in all cells (apart from germ cells), DNA methylation patterns are highly cell- and tissue-specific, and there can be more similarity in DNA methylation patterns between tissues from different species than from different tissues within species [55]. This has important implications when designing DNA methylation studies in general and for ecological epigenetic studies in particular where the possibility to sample tissues can be limited, especially for higher eukaryotes like birds and mammals. I have recently reviewed what we currently understand about how DNA methylation patterns across tissues are reflected in blood samples in ecological epigenetic studies [16] and will cover the main insights from this work here. First, it is important to consider the possibility to sample the tissue which is involved either directly or indirectly in producing the phenotype of interest. For example, the hypothalamus plays a central role in circannual rhythms in mammals, and if one is interested in testing if changes in DNA methylation regulates seasonal phenotypes (e.g. timing of reproduction), then sampling the hypothalamus would be a good starting point, as has been done in Siberian hamsters [56] for example. This of course is not possible in many cases, and the question then arises as to whether there are correlated DNA methylation patterns across tissues, so that DNA methylation patterns would still be informative (e.g. for causing change in expression patterns) for the phenotype studied. This issue has been addressed in humans: Gunasekara *et al.* [57] identified around 10 000 genomic regions that showed consistent DNA methylation patterns across tissues within individuals but that were variable between individuals and thus could potentially explain between-individual differences. It seems reasonable that such correlations between DNA methylation patterns exist, but to what extent this extends across different tissues and to more easily accessible tissue types, such as blood, is still unclear [16]. Another complication is that blood as a tissue consists of a variety of cell types [58], and cell composition can change over the season [59] and differ among species (e.g. in birds, the red blood cells are nucleated unlike in mammals). As each cell type can have its unique DNA methylation profile, cell type variation should be accounted for [40] or, alternatively, single cell types analysed (e.g. [5]). This may not always be feasible of course, but it is important to keep in mind when interpreting results.

## (e) Statistical considerations for identifying differentially methylated sites or regions

Lea *et al.* [40] have recently reviewed different modelling approaches for bisulfite sequencing data. I will summarize the main findings here and also point out one additional consideration not discussed by those authors. As for each CpG site one obtains count information (number of methylated versus unmethylated counts), bisulfite data are best modelled

as a binomial process where the total number of counts at a site models variation in coverage among sites (which can be large particularly in RRBS datasets). However, as there is more variation than expected in a binomial process, beta-binomial models are preferred. A very important point in all studies is the presence of population stratification (population structure and relatedness) that needs to be accounted for in the models. Generalized linear mixed effect models allow one to include a kinship matrix to control for relatedness among individuals, a method familiar to many interested in the genetic basis of phenotypic traits in natural populations [60]. Using this approach is important because DNA methylation shows strong clustering with relatedness, and more closely related individuals will on average show more similar DNA methylation patterns [40].

One additional concern specific to bisulfite data is that many sites in the genome will be completely unmethylated or completely methylated (and which sites these are will vary with tissue type). This poses a potential problem when making statistical inferences; however, as when inspecting the $p$-value histogram, there is typically a large deviation from uniformity with a large peak at the high end of $p$-values which contain sites that do not change in methylation level between treatments or comparison groups. A workaround seems to have developed within the field to circumvent this problem by using a filtering threshold such that sites with less than, say, 10% or 20% difference in methylation are removed prior to statistical testing (e.g. [42,61]). The consequence of such filtering is however not well explored, and it is not unlikely that it may inflate the number of statistically significant tests and thus lead to bias in the number of sites inferred to be statistically significant. This could have the consequence that the relative importance of DNA methylation is overestimated, and future studies that examine and develop methods that can control for the highly skewed $p$-value distribution without *a priori* filtering would be useful.

## 5. What can natural populations contribute to our understanding of epigenetics in evolution?

One might assume that laboratory systems are best suited to study such complex phenomena as epigenetic patterns. However, while laboratory studies offer the ability to carry out highly controlled experiments, the stable and often uniform environments in a laboratory and the fact that both expression of genetic and phenotypic variance (as well as fitness) is strongly determined by the environment [62] mean that the relevance of epigenetic mechanisms and selection operating on them are best studied in the natural environment of the organism. For example, to understand adaptation, we need to understand what determines between-individual variation in a trait, if such variation is heritable, and whether variation in the trait is associated with variation in survival or in reproductive success. This is rarely achieved with epigenetic studies at present, but expanding the genetic studies of long-term studies on wild vertebrates and birds [63] to include also epigenetic information could provide such information and therefore also insights into the role of epigenetics in adaptive

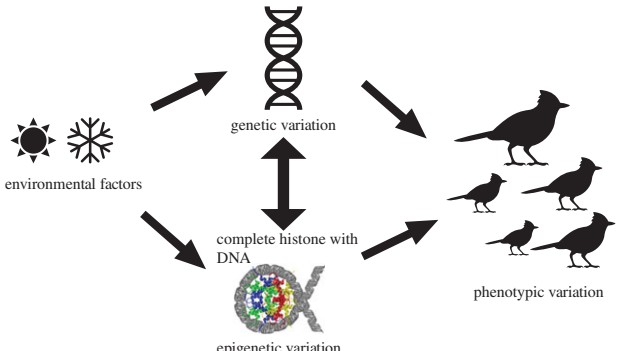

**Figure 1.** Environmental factors modulate both genetic and epigenetic variation which in concert with the environment produce phenotypic variation in the population. (Online version in colour.)

evolution (figure 1). This focus on studying epigenetics in natural populations of course does not negate the value of controlled laboratory studies, which are often better suited to disentangle the contribution epigenetic effects may have on phenotypic variation [17,64], and thus there is much to be gained not least by functional screens in the laboratory. Nevertheless, an understanding of the environmental factors and the selective forces operating on epigenetic variation will be beyond the reach of laboratory studies and need studies on populations in the wild [17].

### (a) Selection on epigenetic marks

Measuring ecologically relevant selection needs to be done in the natural environment of the organism, so studies of natural populations that combine genetic and epigenetic information with phenotypic and fitness data are ideal for providing insights into whether selection acts on epigenetic marks.

Unfortunately, no studies in natural populations have done this yet, to my knowledge, but a recent experiment on recombinant inbred lines in *A. thaliana* under a simulated fragmented landscape compared how patterns of genetic and epigenetic variation responded to selection in controlled environment under a rapidly changing environment [65]. Rapid changes in flowering time and morphology in the selection lines compared to the ancestral control remained for three generations following the completion of selection, and although there was no reduction in genetic diversity observed, there was a clear reduction in epigenetic diversity, with changes in DNA methylation levels, some of which were associated with the observed phenotypic changes [65]. This suggests that epigenetic variation is subject to selection and can be involved in adaptive responses, although how representative the observed changes are for adaptation in natural populations of *A. thaliana* is less clear.

### (b) Transgenerational stability of epigenetic marks

Numerous definitions of epigenetic inheritance are in use, but here I will define it as a situation where epigenetic marks are inherited across generations [25]. Some invoke the need for such epigenetic marks to have a phenotypic effect [32] or that it should be environmentally induced epigenetic marks

that are inherited [66]. These last two definitions I consider a subset of the inheritance of epigenetic marks in general.

One early example of transgenerational epigenetic effects comes from a laboratory study on coat coloration in mice where expression of the methylation-sensitive agouti yellow allele is modulated by the mother's diet and shows epigenetic inheritance [67].

The presence of such transgenerational epigenetic effects and their general importance for evolution is controversial yet is a fascinating aspect of modern genetics subject to much debate [25,31,33,66,68]. Epigenetic studies in natural populations can provide useful insights into this debate. Not only do we have a well-developed theoretical framework for disentangling environmental and genetic effects on phenotypes (the field of quantitative genetics), but careful records of phenotypes (including fitness) and environmental variables can allow us to address various subtleties of the epigenetic inheritance concept, such as whether environmentally induced changes in epigenetic marks with phenotypic effect are inherited across generations and are under selection. In particular, long-term pedigreed studies of natural animal populations provide a rich resource with available blood and/or tissue samples stretching back decades and spanning tens of generations in some cases. Keeping in mind the potential problem of sample storage condition, as well as caveats related to the tissue sampled for epigenetic marks, these long-term pedigreed studies are an excellent opportunity for a detailed analysis of the evolutionary stability of DNA methylation marks in natural populations. Estimates of heritability of DNA methylation marks (epialleles) at individual CpG sites throughout the genome could be generated by combining high-throughput sequencing methods for obtaining epigenetic information (see above) with the linear mixed effects models that the community using long-term studies is familiar with [60]. This would allow variance partitioning of genetic and epigenetic variation and insights into transgenerational stability of epigenetic marks, and also offer a chance to obtain estimates that are (relatively) free of potential genetic confounding effects [69].

## (c) Relative role of epigenetic versus genetic variation

Is epigenetic variation more available than genetic variation for selection to act on and thus contributes to adaptation? This is a relevant question not least in the context of invasive populations, which often have low genetic diversity due to bottlenecks, and where epigenetic variation has been suggested as a potential reservoir for facilitating the success of invasive species [70–72]. In some respects, we expect epigenetic variation to exceed genetic variation as the epigenetic mutation rate can be orders of magnitude higher than the genetic mutation rate [73]. Couple that with the potential for environmentally induced epigenetic variation, the expectation that epigenetic variation should exceed genetic variation is perhaps not surprising.

Several studies have examined population structure using epigenetic as well as genetic data, and at least thus far the evidence is mixed regarding the correlation between genetic and epigenetic variation [71,74,75]. However, just like at the genetic level, significant population differentiation at the epigenetic level has been found [76–80].

## 6. Taking a quantitative genetic approach to integrate epigenetics into studies on wild populations

There are many demonstrations of epigenetic mechanisms generating phenotypic variation in natural populations. However, the extent to which these epigenetic patterns are ultimately controlled by genetic mechanisms is still unclear in many cases, and the potential role that epigenetic may have in long-term evolutionary dynamics (i.e. if epigenetic marks are stable enough to provide an evolutionary response to selection) is contentious. While many studies have examined transgenerational epigenetic effects, disentangling epigenetic variation from genetic variation is challenging and an area of active research, and some of this work may need a revisit following updated methods and experimental designs.

Applying quantitative genetic methods to long-term studies from natural populations to incorporate epigenetic information can allow for accurate estimation of the proportion of phenotypic variance due to additive genetic effects and that due to epigenetic effects [81]. Like other genome-wide methods for identifying genetic loci contributing to a trait (GWAS), the use of epigenetic data will similarly allow for identification of epigenetic loci that contribute to the phenotype (EWAS) while controlling for genetic effects. However, this will require large sample sizes in terms of both number of individuals and number of generations sampled to obtain unbiased estimates [82], as well as dense sampling in terms of number of genetic loci (SNPS, SVs) and epigenetic loci to achieve statistical power to detect loci that are likely to have small effects. This integration of 'molecular phenotypes' in a quantitative genetic framework combined with fitness data to estimate selection [83] will make it possible to predict and better understand phenotypic evolution and the relative role of genetic and epigenetic factors.

We are currently in the midst of a flurry of correlative studies that probe the potential role of epigenetic mechanisms on natural populations. The challenge will be to move beyond this correlative evidence and rule out potential confounding genetic effects in order to fully understand the importance and generality of epigenetic mechanisms in evolutionary dynamics in natural populations. Larger and more encompassing studies will help with more robust inferences, and we are also starting to see functional studies targeting also epigenetic mechanisms performed on natural populations (e.g. [22,30]), which allow for unequivocal evidence of the role of epigenetics in phenotypic evolution.

Data accessibility. This article has no additional data.

Competing interests. I declare I have no competing interests.

Funding. The Norwegian Research Council (grant nos. 239974 and 223257) is acknowledged for funding.

Acknowledgements. I would like to thank the editors for inviting me to contribute to this special issue and for their patience during the revision stages, as well as two anonymous reviewers for helpful comments and suggestions.

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
