## [Peer Review File · Proceedings of the Royal Society B: Biological Sciences]

Review History

RSPB-2020-2838.R0 (Original submission)

Review form: Reviewer 1

Recommendation

Major revision is needed (please make suggestions in comments)

Scientific importance: Is the manuscript an original and important contribution to its field?

Acceptable

General interest: Is the paper of sufficient general interest?

Good

Quality of the paper: Is the overall quality of the paper suitable?

Marginal

Is the length of the paper justified?

Yes

Should the paper be seen by a specialist statistical reviewer?

No

Do you have any concerns about statistical analyses in this paper? If so, please specify them explicitly in your report.

No

It is a condition of publication that authors make their supporting data, code and materials available - either as supplementary material or hosted in an external repository. Please rate, if applicable, the supporting data on the following criteria.

Is it accessible?

N/A

Is it clear?

N/A

Is it adequate?

N/A

Do you have any ethical concerns with this paper?

No

Comments to the Author

This review paper provides an overview of current work investigating the role of epigenetic mechanisms in generating phenotypic diversity in natural populations. My general impression is that the review does a fairly good job at explaining the important issues, but it does so by reiterating the same points that have been made in a number of previous reviews and perspective papers (e.g., Richards 2006, Bossdorf et al. 2008, Richards et al. 2010, Danchin et al. 2011, Shea et al. 2011, Verhoeven et al. 2016, Hu and Barrett 2017 – note that not all of these papers are cited). This review also uses many of the same case studies as the previous papers in order to explain these concepts. As such, although this is an interesting topic and the paper will be relevant to the readership of this journal, I don't feel the present review adds a great deal of novel insight to the existing literature. In addition, there are several relevant papers that are not covered here. Although this is a rapidly expanding field, the number of studies investigating epigenetics in natural populations is still quite limited, and as such it would be good to be as comprehensive as possible in covering existing work. Below my comments I have listed some studies that would be good to include in a review on this topic if space permits.

I feel that the strongest element of the paper is the discussion on the strengths and weaknesses of different methodological approaches for conducting epigenetics with natural populations. This is an area that has not been covered in as much detail in previous work and will be very useful for molecular ecologists that are new to epigenetics and need to make practical decisions about study design and methodology. Methods in this field are also moving very quickly such that any discussion of sequencing approaches in papers from even just a few years ago begins to lose relevance.

Below I provide some comments that I hope will be useful.

General: A central challenge that is highlighted in the review is the need to identify the role of epigenetic mechanisms operating independently of genetic variation. However, although it is stated repeatedly that studying natural populations will be important for achieving this goal, I don't feel the paper really articulates why. If anything, one would expect that this precise challenge would best be addressed by careful studies under controlled lab conditions, which could best disentangle environmental, genetic, and epigenetic effects on phenotypic variation. Of course, the ecological relevance of the environmental factors tested in the lab may be unclear, and the genetic backgrounds and epigenetic variation present in lab-raised organisms are often not representative of natural populations. But this might not matter if the only goal is to obtain a

mechanistic understanding of the interactions between the environment, genetic variation, and epigenetic variation. This is not to say that studies in natural populations are not needed – they clearly are. But I don't think this paper makes a convincing case why. What are the specific situations in which natural populations provide novel insights into epigenetic mechanisms that we cannot attain through lab studies? It seems important for a review on this topic to clearly articulate this.

General: I'm afraid the paper is not very well-written, with unclear wording in many sections, repetitive sentences, and numerous grammar mistakes and typos that sometimes impede interpretation of the text. I have done my best to flag these. A revised manuscript would benefit from a thorough re-write to improve the clarity and structure of the paper.

General: It might be useful to have a table summarizing the characteristics of the various methods described in the paper. There are no display items of any kind in the paper, which is unusual. For instance, it might also be nice to include a figure showing a key result from one of the empirical papers that is cited to help illustrate the point being made. Or some kind of conceptual diagram to help demonstrate how genetic and epigenetic mechanisms could independently contribute to phenotypic variation, etc.

Section 1. "The emerging role of epigenetic mechanisms in generating phenotypic diversity": I don't think this section actually tells us anything about the role of epigenetic mechanisms in generating phenotypic diversity. It describes how work in this area is increasing and outlines some methodological advances but doesn't explain what they have revealed. It also seems like it would be useful to provide some basic definitions of epigenetic terms/mechanisms in this section and explain how they can generate phenotypic diversity. E.g., how do histone modifications or small RNAs change phenotype? This will be helpful for newcomers to epigenetics.

L19: Use of "also" is unclear. Presumably this is meant to convey that epigenetic mechanisms are now increasingly studied in natural populations as well as lab-reared organisms, but coming without context in the first sentence of the abstract this is not clear.

L24: "much of this is" -> "many of these studies are"

L25: "contribution" -> "contributions"

L27: "resolve" -> "resolving"

L28: Incorrect grammar.

L37: "#REF". Please add the citations.

L39: "at" -> "with"

L41: "mid 2000" -> "the mid 2000s"

L44: Here and in several other places in the paper the use of "also" is unclear. It would be good to be more precise about what is being compared with. Presumably model species is meant here, but the wording is vague (e.g., this could also be drawing a contrast with scales lower than genome-wide).

L50: "natural population" -> "natural populations"

L51: "techniques" -> "technique"

L51: methylation spelled wrong.

L53: "advancement" -> "advancements"

L64: Typo, Incorrect grammar.

L71: Perhaps define 'peloric'.

L73: The publication of a single paper doesn't necessarily put something "firmly on the map". How did it change opinions? Was it well-cited? Did many other papers on the topic soon follow?

L79: "seem to contribute". But to what extent? Here and in other places it would be useful to provide a bit more nuance and insight about the relative roles of different mechanisms. Few people would dispute that epigenetic mechanisms can contribute in some way to phenotypic variation. The more interesting question (and one that would hopefully be addressed in a review on this specific topic) is whether it is a meaningful contribution. Or if the relative roles of epigenetic vs. genetic variation shift under particular ecological or evolutionary circumstances.

L83: "sate" -> "state".

L92: "plants" -> "plant"

L109: "here studies on natural populations can play a key role". Again, it is unclear why natural populations are needed for addressing issues this specific issue. In many ways I would expect a controlled lab study to be best suited for isolating the impact of transgenerational epigenetic effects on phenotypic variation.

L121: The bisulphite conversion is not specific to reduced representation approaches. This could be moved to more general section of text.

L141: "require" -> "requires"

L145: Define "epiGBS". There is inconsistency about which terms/acronyms are defined and which are not (e.g., RRBS is, but epiGBS, ddRAD, and ATAC-seq are not).

L150: It might be good to explain why interest in CpG methylation would be associated with study system.

L156: For consistency, state whether epiRAD requires a reference genome.

L158: It is not ideal to have a single sentence paragraphs like this (here and elsewhere in the paper). Perhaps try to merge them with other text.

L167: "offer" -> "offers"

L168: It would be nice to provide a little more explanation about how different types of epigenetic mechanisms (e.g., CpG methylation vs. non-CpG methylation, etc.) vary and what the ecological/evolutionary consequences of this variation could be.

L172: The context for this result is unclear. Is this in a single individual? What is the generality?

L198: "is" -> "are". Might be nice to list what these projects are.

L206: Explain why less input material is required.

L208: "control" -> "controls"

L211: “even have consequences for population demography”. This is unclear. In addition to what?

L213: This same example was already given and described earlier in the paper.

L216: Explain what RNAi is.

L225: This is a bit repetitive with what has come earlier.

L226: A single individual could potentially be used, but the generality of the inferences would be limited.

L234: “reduced” -> “reduced representation”.

L241: Incorrect grammar.

L244: These two citations don’t necessarily indicate those fragment sizes are “commonly” used in birds or mammals – only a single species is used in each of the papers.

L255: Incorrect grammar.

L258: “receptive” -> “repetitive”

L260: Incorrect grammar.

L265: Other enzymes could also be used.

L321: “typically” and “most commonly” are redundant.

L334: “play” -> “plays”

L344: “exists” -> “exist”

L346: “consist” -> “consists”

L356: Incorrect grammar.

L365: Incorrect grammar.

L367: “show” -> “shows”

L370: “(always unmethylated)”. My understanding is that hypomethylation and hypermethylation of DNA are relative terms and denote less or more methylation when compared to some DNA standard, so use of hypomethylation would not necessarily indicate that a site is always unmethylated (at least at a population level).

L375: “p-value” -> “p-values”

L379: I might not be thinking about this clearly, but the logic here seems backwards. Wouldn’t leaving the uninformative sites in the dataset be more likely to result in test statistic inflation? Excluding these sites prior to statistical testing seems like a sensible approach to avoid false positives for significant hyper/hypo-methylation. If they are left in the dataset, they will indeed reduce the proportion of sites that show a change in methylation since the denominator will now include a large number of sites showing no change. But, these sites will also reduce the genome-wide threshold for significance to make it easier for any individual site that does show a change in methylation to test as being significantly differentially methylated. I think most researchers are

more interested in confidently identifying large and meaningful changes in methylation, as opposed to knowing what proportion of sites show a change vs. no change (since the magnitude of many of the changes will be small and therefore will not have meaningful impacts). As an analogy, one would not want to identify statistically significant *Fst* outliers using a dataset that contains uniformly invariant sites.

L390: Measuring selection does not need to be done in the natural environment of the organism. This is required for measuring ecologically-relevant natural selection, but not 'selection'.

L391: And phenotypic data as well, presumably.

L393: This example directly contradicts the opening point made for this section. The *A. thaliana* experiment was done under controlled conditions, not natural conditions.

L394: Incorrect grammar.

L410: Why do you find them the most interesting?

L426: "induces" -> "induced"

L429: Citations?

L441: "for" -> "of"

L443: It would be nice to explain how this is accomplished.

L448: Wording could be improved.

L452: Incorrect grammar/wording.

L454: It would be useful to explain this a bit more for readers who are not familiar with the resetting of epigenetic marks. E.g., what proportion is typically re-set? Does this vary across groups? Does it vary based on environmental conditions?

L459: This is a bit vacuous. It could potentially contribute to local adaptation or it could not. It would depend on a number of factors that aren't really explained.

L468: Incorrect grammar.

L472: Remove "to".

L475: "mechanism" -> "mechanisms"

References: A mix of formatting styles are used and there are spelling mistakes throughout.

Below are some relevant studies on epigenetics in natural populations or under ecologically relevant conditions that are not cited. These are just papers that I remember and is not an exhaustive list. But given this is a formal review paper, it would be good to cover as much of the relevant literature as possible given this is a fairly new field and there are still not very many data papers on the topic.

McNew SM, Beck D, Sadler-Riggelman I, Knutie SA, Koop JAH, Clayton DH, Skinner MK. 2017. Epigenetic variation between urban and rural populations of Darwin's finches. *BMC Evol Biol.* 17:183.

- Baerwald MR, Meek MH, Stephens MR, Nagarajan RP, Goodbla AM, Tomalty KM, Thorgaard GH, May B, Nichols KM. 2016. Migration-related phenotypic divergence is associated with epigenetic modifications in rainbow trout. *Mol Ecol.* 25(8):1785–1800.
- Le Luyer J, Laporte M, Beacham TD, Kaukinen KH, Withler RE, Leong JS, Rondeau EB, Koop BF, Bernatchez L. 2017. Parallel epigenetic modifications induced by hatchery rearing in a Pacific salmon. *Proc Natl Acad Sci U S A.* 114(49):12964–12969.
- Hu J, Askary AM, Thurman TJ, Spiller D, Palmer TM, Pringle RM, Barrett RDH. 2019. Epigenetic signatures of colonizing new environments in *Anolis* lizards. *Mol Biol Evol* 36(10): 2165–2170.
- Huang X, Li S, Ni P, Gao Y, Jiang B, Zhou Z, Zhan A. 2017. Rapid response to changing environments during biological invasions: DNA methylation perspectives. *Mol Ecol.* 26(23):6621–6633.
- Uren Webster TM, Rodriguez-Barreto D, Martin SAM, van Oosterhout C, Wengel P, Cable J, Hamilton A, Garcia de Leaniz C, Consuegra S. 2018. Contrasting effects of acute and chronic stress on the transcriptome, epigenome, and immune response of Atlantic salmon. *Epigenetics* 13(12):1191–1207.
- Liebl, A.L., Schrey, A.W., Richards, C.L. & Martin, L.B. 2013. Patterns of DNA methylation throughout a range expansion of an introduced songbird. *Integr. Comp. Biol.* 53: 351–358.
- Liu, S., Sun, K., Jiang, T., Ho, J.P., Liu, B. & Feng, J. 2012. Natural epigenetic variation in the female great roundleaf bat *Hipposideros armiger* populations. *Mol. Genet. Genomics* 287: 643–650.
- Blouin, M.S., Thuilier, V., Cooper, B., Amarasinghe, V., Cluzel, L., Hitoshi, A. et al. 2010. No evidence for large differences in genomic methylation between wild and hatchery steelhead *Oncorhynchus mykiss*. *Can. J. Fish Aquat. Sci.* 67: 217–224.
- Dimond, J.L. & Roberts, S.B. 2016. Germline DNA methylation in reef corals: patterns and potential roles in response to environmental change. *Mol. Ecol.* 25: 1895–1904.
- Dubin, M.J., Zhang, P., Meng, D., Remigereau, M.S., Osborne, E.J., Paolo, C.F. et al. 2015. DNA methylation in *Arabidopsis* has a genetic basis and shows evidence of local adaptation. *eLife* 4: e05255.
- Foust, C.M., Preite, V., Schrey, A.W., Alvarez, M., Robertson, M.H., Verhoeven, K.J.F. et al. 2016. Genetic and epigenetic differences associated with environmental gradients in replicate populations of two salt marsh perennials. *Mol. Ecol.* 25: 1639–1652.
- Gao, L., Geng, Y., Li, B., Chen, J. & Yang, J. 2010. Genome-wide DNA methylation alterations of *Alternanthera philoxeroides* in natural and manipulated habitats: implications for epigenetic regulation of rapid responses to environmental fluctuation and phenotypic variation. *Plant, Cell Environ.* 33: 1820–1827.
- Gugger, P., Fitz-Gibbon, S., Pellegrini, M. & Sork, V.L. 2016. Species-wide patterns of DNA methylation variation in *Quercus lobata* and its association with climate gradients. *Mol. Ecol.* 25: 1665–1680.
- Keller, T.E., Lasky, J.R. & Yi, S.V. 2016. The multivariate association between genome-wide DNA methylation and climate across the range of *Arabidopsis thaliana*. *Mol. Ecol.* 25: 1823–1837.
- Lea, A.J., Altmann, J., Alberts, S.C. & Tung, J. 2016. Resource base influences genome-wide DNA methylation levels in wild baboons *Papio cynocephalus*. *Mol. Ecol.* 25: 1681–1696

Verhoeven, K.J.F., Jansen, J.J., van Dijk, P.J. & Biere, A. 2010. Stress-induced DNA methylation changes and their heritability in asexual dandelions. *New Phytol.* 185: 1108– 1118.

Review form: Reviewer 2

Recommendation

Accept with minor revision (please list in comments)

Scientific importance: Is the manuscript an original and important contribution to its field?

Good

General interest: Is the paper of sufficient general interest?

Good

Quality of the paper: Is the overall quality of the paper suitable?

Good

Is the length of the paper justified?

Yes

Should the paper be seen by a specialist statistical reviewer?

No

Do you have any concerns about statistical analyses in this paper? If so, please specify them explicitly in your report.

No

It is a condition of publication that authors make their supporting data, code and materials available - either as supplementary material or hosted in an external repository. Please rate, if applicable, the supporting data on the following criteria.

Is it accessible?

N/A

Is it clear?

N/A

Is it adequate?

N/A

Do you have any ethical concerns with this paper?

No

Comments to the Author

Husby reviewed the recent progress of the epigenetics studies in plants and animals. The authors discussed the emerging role of epigenetic variation, some technical aspects, and potential analyses that can be done. Epigenetic variation, especially from the evolutionary perspective, is a timely and hot topic. I believe it will be of great interest to a broader audience of the journal. The paper is well organized and is easy to follow. If the author can make some minor revisions and cite some recently published relevant publications, I think it is publishable. See below my detailed comments.

- 1) I strongly believe a figure or diagram is needed to recapitulate the review's major contents. For example, list the pros and cons of the different approaches for measuring DNA methylation in natural populations would be helpful.
- 2) I'd appreciate it if the author can generalize the epigenetic measurement section (from line 112) to include not only animals but plants.
- 3) Some recent publications highly relevant to the current review should consider being cited.

Shahryary, Y., Symeonidi, A., Hazarika, R.R. et al. AlphaBeta: computational inference of epimutation rates and spectra from high-throughput DNA methylation data in plants. *Genome Biol* 21, 260 (2020). <https://doi.org/10.1186/s13059-020-02161-6>

Xu, G., Lyu, J., Li, Q. et al. Evolutionary and functional genomics of DNA methylation in maize domestication and improvement. *Nat Commun* 11, 5539 (2020). <https://doi.org/10.1038/s41467-020-19333-4>

Decision letter (RSPB-2020-2838.R0)

17-Dec-2020

Dear Professor Husby:

I am writing to inform you that your manuscript RSPB-2020-2838 entitled "Wild epigenetics: insights from epigenetic studies on natural populations" has, in its current form, been rejected for publication in *Proceedings B*. Below you will find full referee comments, and the recommendation made by the board member. In addition, I would strongly urge you, in any resubmission, that you include in a more explicit fashion, how and why this particular manuscript, advances the field, especially in relation to perceived novelty. That element is critical, to our consideration for onward processing of your manuscript. There are various reviews that already exist, and for whatever reason have not been sufficiently covered (detailed by referee #1), and it is vital that in very clear terms, you include a full justification for your manuscript, and how it advances our understanding significantly beyond the current state of art.

This action has been taken on the advice of referees, who have recommended that substantial revisions are necessary. With this in mind we would be happy to consider a resubmission, provided the comments of the referees are fully addressed. However please note that this is not a provisional acceptance.

4) Data - please see our policies on data sharing to ensure that you are complying (<https://royalsociety.org/journals/authors/author-guidelines/#data>).

Sincerely,
Professor Gary Carvalho
mailto: proceedingsb@royalsociety.org

Associate Editor
Board Member: 1
Comments to Author:
Dear Dr. Husby

Thank you for your submission to our special issue on Wild Quantitative Genetics in Proc R Soc B. Your manuscript has now been reviewed by two experts in the field. Both reviewers were positive about some aspects of the manuscript, especially its discussion of the strengths and weaknesses of different methods, which they thought would be even more strong with a figure or table. However, one of the reviewers had a number of critiques that should be addressed. In particular, I agree that further elaborating the importance of studying epigenetics in wild populations and describing how this could be accomplished with a quantitative genetics approach would improve the manuscript.

Reviewer(s)' Comments to Author:

Referee: 1

Comments to the Author(s)

This review paper provides an overview of current work investigating the role of epigenetic mechanisms in generating phenotypic diversity in natural populations. My general impression is that the review does a fairly good job at explaining the important issues, but it does so by reiterating the same points that have been made in a number of previous reviews and perspective papers (e.g., Richards 2006, Bossdorf et al. 2008, Richards et al. 2010, Danchin et al. 2011, Shea et al. 2011, Verhoeven et al. 2016, Hu and Barrett 2017 - note that not all of these papers are cited). This review also uses many of the same case studies as the previous papers in order to explain these concepts. As such, although this is an interesting topic and the paper will be relevant to the readership of this journal, I don't feel the present review adds a great deal of novel insight to the existing literature. In addition, there are several relevant papers that are not covered here. Although this is a rapidly expanding field, the number of studies investigating epigenetics in natural populations is still quite limited, and as such it would be good to be as comprehensive as possible in covering existing work. Below my comments I have listed some studies that would be good to include in a review on this topic if space permits.

I feel that the strongest element of the paper is the discussion on the strengths and weaknesses of different methodological approaches for conducting epigenetics with natural populations. This is an area that has not been covered in as much detail in previous work and will be very useful for molecular ecologists that are new to epigenetics and need to make practical decisions about study design and methodology. Methods in this field are also moving very quickly such that any discussion of sequencing approaches in papers from even just a few years ago begins to lose relevance.

Below I provide some comments that I hope will be useful.

General: A central challenge that is highlighted in the review is the need to identify the role of epigenetic mechanisms operating independently of genetic variation. However, although it is

stated repeatedly that studying natural populations will be important for achieving this goal, I don't feel the paper really articulates why. If anything, one would expect that this precise challenge would best be addressed by careful studies under controlled lab conditions, which could best disentangle environmental, genetic, and epigenetic effects on phenotypic variation. Of course, the ecological relevance of the environmental factors tested in the lab may be unclear, and the genetic backgrounds and epigenetic variation present in lab-raised organisms are often not representative of natural populations. But this might not matter if the only goal is to obtain a mechanistic understanding of the interactions between the environment, genetic variation, and epigenetic variation. This is not to say that studies in natural populations are not needed – they clearly are. But I don't think this paper makes a convincing case why. What are the specific situations in which natural populations provide novel insights into epigenetic mechanisms that we cannot attain through lab studies? It seems important for a review on this topic to clearly articulate this.

General: I'm afraid the paper is not very well-written, with unclear wording in many sections, repetitive sentences, and numerous grammar mistakes and typos that sometimes impede interpretation of the text. I have done my best to flag these. A revised manuscript would benefit from a thorough re-write to improve the clarity and structure of the paper.

General: It might be useful to have a table summarizing the characteristics of the various methods described in the paper. There are no display items of any kind in the paper, which is unusual. For instance, it might also be nice to include a figure showing a key result from one of the empirical papers that is cited to help illustrate the point being made. Or some kind of conceptual diagram to help demonstrate how genetic and epigenetic mechanisms could independently contribute to phenotypic variation, etc.

Section 1. "The emerging role of epigenetic mechanisms in generating phenotypic diversity": I don't think this section actually tells us anything about the role of epigenetic mechanisms in generating phenotypic diversity. It describes how work in this area is increasing and outlines some methodological advances but doesn't explain what they have revealed. It also seems like it would be useful to provide some basic definitions of epigenetic terms/mechanisms in this section and explain how they can generate phenotypic diversity. E.g., how do histone modifications or small RNAs change phenotype? This will be helpful for newcomers to epigenetics.

L19: Use of "also" is unclear. Presumably this is meant to convey that epigenetic mechanisms are now increasingly studied in natural populations as well as lab-reared organisms, but coming without context in the first sentence of the abstract this is not clear.

L24: "much of this is" -> "many of these studies are"

L25: "contribution" -> "contributions"

L27: "resolve" -> "resolving"

L28: Incorrect grammar.

L37: "#REF". Please add the citations.

L39: "at" -> "with"

L41: "mid 2000" -> "the mid 2000s"

L44: Here and in several other places in the paper the use of "also" is unclear. It would be good to be more precise about what is being compared with. Presumably model species is meant here, but the wording is vague (e.g., this could also be drawing a contrast with scales lower than genome-wide).

L50: "natural population" -> "natural populations"

L51: "techniques" -> "technique"

L51: methylation spelled wrong.

L53: "advancement" -> "advancements"

L64: Typo, Incorrect grammar.

L71: Perhaps define 'peloric'.

L73: The publication of a single paper doesn't necessarily put something "firmly on the map". How did it change opinions? Was it well-cited? Did many other papers on the topic soon follow?

L79: "seem to contribute". But to what extent? Here and in other places it would be useful to provide a bit more nuance and insight about the relative roles of different mechanisms. Few people would dispute that epigenetic mechanisms can contribute in some way to phenotypic variation. The more interesting question (and one that would hopefully be addressed in a review on this specific topic) is whether it is a meaningful contribution. Or if the relative roles of epigenetic vs. genetic variation shift under particular ecological or evolutionary circumstances.

L83: "sate" -> "state".

L92: "plants" -> "plant"

L109: "here studies on natural populations can play a key role". Again, it is unclear why natural populations are needed for addressing issues this specific issue. In many ways I would expect a controlled lab study to be best suited for isolating the impact of transgenerational epigenetic effects on phenotypic variation.

L121: The bisulphite conversion is not specific to reduced representation approaches. This could be moved to more general section of text.

L141: "require" -> "requires"

L145: Define "epiGBS". There is inconsistency about which terms/acronyms are defined and which are not (e.g., RRBS is, but epiGBS, ddRAD, and ATAC-seq are not).

L150: It might be good to explain why interest in CpG methylation would be associated with study system.

L156: For consistency, state whether epiRAD requires a reference genome.

L158: It is not ideal to have a single sentence paragraphs like this (here and elsewhere in the paper). Perhaps try to merge them with other text.

L167: "offer" -> "offers"

L168: It would be nice to provide a little more explanation about how different types of epigenetic mechanisms (e.g., CpG methylation vs. non-CpG methylation, etc.) vary and what the ecological/evolutionary consequences of this variation could be.

L172: The context for this result is unclear. Is this in a single individual? What is the generality?

L198: "is" -> "are". Might be nice to list what these projects are.

L206: Explain why less input material is required.

L208: "control" -> "controls"

L211: "even have consequences for population demography". This is unclear. In addition to what?

L213: This same example was already given and described earlier in the paper.

L216: Explain what RNAi is.

L225: This is a bit repetitive with what has come earlier.

L226: A single individual could potentially be used, but the generality of the inferences would be limited.

L234: "reduced" -> "reduced representation".

L241: Incorrect grammar.

L244: These two citations don't necessarily indicate those fragment sizes are "commonly" used in birds or mammals - only a single species is used in each of the papers.

L255: Incorrect grammar.

L258: "receptive" -> "repetitive"

L260: Incorrect grammar.

L265: Other enzymes could also be used.

L321: "typically" and "most commonly" are redundant.

L334: "play" -> "plays"

L344: "exists" -> "exist"

L346: "consist" -> "consists"

L356: Incorrect grammar.

L365: Incorrect grammar.

L367: "show" -> "shows"

L370: "(always unmethylated)". My understanding is that hypomethylation and hypermethylation of DNA are relative terms and denote less or more methylation when compared to some DNA standard, so use of hypomethylation would not necessarily indicate that a site is always unmethylated (at least at a population level).

L375: "p-value" -> "p-values"

L379: I might not be thinking about this clearly, but the logic here seems backwards. Wouldn't leaving the uninformative sites in the dataset be more likely to result in test statistic inflation?

Excluding these sites prior to statistical testing seems like a sensible approach to avoid false positives for significant hyper/hypo-methylation. If they are left in the dataset, they will indeed reduce the proportion of sites that show a change in methylation since the denominator will now include a large number of sites showing no change. But, these sites will also reduce the genome-wide threshold for significance to make it easier for any individual site that does show a change in methylation to test as being significantly differentially methylated. I think most researchers are more interested in confidently identifying large and meaningful changes in methylation, as opposed to knowing what proportion of sites show a change vs. no change (since the magnitude of many of the changes will be small and therefore will not have meaningful impacts). As an analogy, one would not want to identify statistically significant *Fst* outliers using a dataset that contains uniformly invariant sites.

L390: Measuring selection does not need to be done in the natural environment of the organism. This is required for measuring ecologically-relevant natural selection, but not 'selection'.

L391: And phenotypic data as well, presumably.

L393: This example directly contradicts the opening point made for this section. The *A. thaliana* experiment was done under controlled conditions, not natural conditions.

L394: Incorrect grammar.

L410: Why do you find them the most interesting?

L426: "induces" -> "induced"

L429: Citations?

L441: "for" -> "of"

L443: It would be nice to explain how this is accomplished.

L448: Wording could be improved.

L452: Incorrect grammar/wording.

L454: It would be useful to explain this a bit more for readers who are not familiar with the resetting of epigenetic marks. E.g., what proportion is typically re-set? Does this vary across groups? Does it vary based on environmental conditions?

L459: This is a bit vacuous. It could potentially contribute to local adaptation or it could not. It would depend on a number of factors that aren't really explained.

L468: Incorrect grammar.

L472: Remove "to".

L475: "mechanism" -> "mechanisms"

References: A mix of formatting styles are used and there are spelling mistakes throughout.

Below are some relevant studies on epigenetics in natural populations or under ecologically relevant conditions that are not cited. These are just papers that I remember and is not an exhaustive list. But given this is a formal review paper, it would be good to cover as much of the relevant literature as possible given this is a fairly new field and there are still not very many data papers on the topic.

- McNew SM, Beck D, Sadler-Riggelman I, Knutie SA, Koop JAH, Clayton DH, Skinner MK. 2017. Epigenetic variation between urban and rural populations of Darwin's finches. *BMC Evol Biol.* 17:183.
- Baerwald MR, Meek MH, Stephens MR, Nagarajan RP, Goodbla AM, Tomalty KM, Thorgaard GH, May B, Nichols KM. 2016. Migration-related phenotypic divergence is associated with epigenetic modifications in rainbow trout. *Mol Ecol.* 25(8):1785–1800.
- Le Luyer J, Laporte M, Beacham TD, Kaukinen KH, Withler RE, Leong JS, Rondeau EB, Koop BF, Bernatchez L. 2017. Parallel epigenetic modifications induced by hatchery rearing in a Pacific salmon. *Proc Natl Acad Sci U S A.* 114(49):12964–12969.
- Hu J, Askary AM, Thurman TJ, Spiller D, Palmer TM, Pringle RM, Barrett RDH. 2019 Epigenetic signatures of colonizing new environments in *Anolis* lizards. *Mol Biol Evol* 36(10): 2165-2170.
- Huang X, Li S, Ni P, Gao Y, Jiang B, Zhou Z, Zhan A. 2017. Rapid response to changing environments during biological invasions: DNA methylation perspectives. *Mol Ecol.* 26(23):6621–6633.
- Uren Webster TM, Rodriguez-Barreto D, Martin SAM, van Oosterhout C, Wengel P, Cable J, Hamilton A, Garcia de Leaniz C, Consuegra S. 2018. Contrasting effects of acute and chronic stress on the transcriptome, epigenome, and immune response of Atlantic salmon. *Epigenetics* 13(12):1191–1207.
- Liebl, A.L., Schrey, A.W., Richards, C.L. & Martin, L.B. 2013. Patterns of DNA methylation throughout a range expansion of an introduced songbird. *Integr. Comp. Biol.* 53: 351–358.
- Liu, S., Sun, K., Jiang, T., Ho, J.P., Liu, B. & Feng, J. 2012. Natural epigenetic variation in the female great roundleaf bat *Hipposideros armiger* populations. *Mol. Genet. Genomics* 287: 643–650.
- Blouin, M.S., Thuilier, V., Cooper, B., Amarasinghe, V., Cluzel, L., Hitoshi, A. et al. 2010. No evidence for large differences in genomic methylation between wild and hatchery steelhead *Oncorhynchus mykiss*. *Can. J. Fish Aquat. Sci.* 67: 217–224.
- Dimond, J.L. & Roberts, S.B. 2016. Germline DNA methylation in reef corals: patterns and potential roles in response to environmental change. *Mol. Ecol.* 25: 1895–1904.
- Dubin, M.J., Zhang, P., Meng, D., Remigereau, M.S., Osborne, E.J., Paolo, C.F. et al. 2015. DNA methylation in *Arabidopsis* has a genetic basis and shows evidence of local adaptation. *eLife* 4: e05255.
- Foust, C.M., Preite, V., Schrey, A.W., Alvarez, M., Robertson, M.H., Verhoeven, K.J.F. et al. 2016. Genetic and epigenetic differences associated with environmental gradients in replicate populations of two salt marsh perennials. *Mol. Ecol.* 25: 1639–1652.
- Gao, L., Geng, Y., Li, B., Chen, J. & Yang, J. 2010. Genome-wide DNA methylation alterations of *Alternanthera philoxeroides* in natural and manipulated habitats: implications for epigenetic regulation of rapid responses to environmental fluctuation and phenotypic variation. *Plant, Cell Environ.* 33: 1820–1827.
- Gugger, P., Fitz-Gibbon, S., Pellegrini, M. & Sork, V.L. 2016. Species-wide patterns of DNA methylation variation in *Quercus lobata* and its association with climate gradients. *Mol. Ecol.* 25: 1665–1680.
- Keller, T.E., Lasky, J.R. & Yi, S.V. 2016. The multivariate association between genome-wide DNA methylation and climate across the range of *Arabidopsis thaliana*. *Mol. Ecol.* 25: 1823–1837.
- Lea, A.J., Altmann, J., Alberts, S.C. & Tung, J. 2016. Resource base influences genome-wide DNA methylation levels in wild baboons *Papio cynocephalus*. *Mol. Ecol.* 25: 1681–1696
- Verhoeven, K.J.F., Jansen, J.J., van Dijk, P.J. & Biere, A. 2010. Stress-induced DNA methylation changes and their heritability in asexual dandelions. *New Phytol.* 185: 1108–1118.

Referee: 2

Comments to the Author(s)

Husby reviewed the recent progress of the epigenetics studies in plants and animals. The authors discussed the emerging role of epigenetic variation, some technical aspects, and potential analyses that can be done. Epigenetic variation, especially from the evolutionary perspective, is a timely and hot topic. I believe it will be of great interest to a broader audience of the journal. The

paper is well organized and is easy to follow. If the author can make some minor revisions and cite some recently published relevant publications, I think it is publishable. See below my detailed comments.

- 1) I strongly believe a figure or diagram is needed to recapitulate the review's major contents. For example, list the pros and cons of the different approaches for measuring DNA methylation in natural populations would be helpful.
- 2) I'd appreciate it if the author can generalize the epigenetic measurement section (from line 112) to include not only animals but plants.
- 3) Some recent publications highly relevant to the current review should consider being cited.

Shahryary, Y., Symeonidi, A., Hazarika, R.R. et al. AlphaBeta: computational inference of epimutation rates and spectra from high-throughput DNA methylation data in plants. *Genome Biol* 21, 260 (2020). <https://doi.org/10.1186/s13059-020-02161-6>

Xu, G., Lyu, J., Li, Q. et al. Evolutionary and functional genomics of DNA methylation in maize domestication and improvement. *Nat Commun* 11, 5539 (2020). <https://doi.org/10.1038/s41467-020-19333-4>

Author's Response to Decision Letter for (RSPB-2020-2838.R0)

See Appendix A.

RSPB-2021-1633.R0

Review form: Reviewer 1

Recommendation

Accept with minor revision (please list in comments)

Scientific importance: Is the manuscript an original and important contribution to its field?

Marginal

General interest: Is the paper of sufficient general interest?

Good

Quality of the paper: Is the overall quality of the paper suitable?

Acceptable

Is the length of the paper justified?

Yes

Should the paper be seen by a specialist statistical reviewer?

No

Do you have any concerns about statistical analyses in this paper? If so, please specify them explicitly in your report.

Yes

It is a condition of publication that authors make their supporting data, code and materials available - either as supplementary material or hosted in an external repository. Please rate, if applicable, the supporting data on the following criteria.

Is it accessible?

N/A

Is it clear?

N/A

Is it adequate?

N/A

Do you have any ethical concerns with this paper?

No

Comments to the Author

I was previously reviewer 1 for this manuscript. I feel that the revisions have addressed a number of my previous concerns. I have a few additional comments:

1. While the addition of two figures helps address the lack of display elements in the manuscript, the two which have been added are relatively superficial: one which provides photos of organisms that have been the focus of epigenetic study and one providing a very basic schematic of interactions between the environment, genetics, epigenetics, and phenotype. These are nice, but I still feel it would be useful to add something more substantive (e.g., a table covering the specific strengths and weaknesses of different approaches mentioned, or their best areas of application, or a figure showing data from one of the key papers cited that can clearly convey one of the central points being made in the text).

2. The central critique in my initial review was that the review did not do an adequate job of articulating precisely how studying natural populations will help to understand the role of epigenetic mechanisms in generating phenotypic diversity. The revised section 5 now briefly addresses this issue by reiterating the point I made in my review, that “the ecological relevance of the environmental factors tested in the lab may be unclear, and the genetic backgrounds and epigenetic variation present in lab-raised organisms are often not representative of natural populations.”, but, as I stated previously, this might not matter if the only goal is to obtain a mechanistic understanding of the interactions between the environment, genetic variation, epigenetic variation, and phenotype. We already have a difficult time disentangling genetic and environmental effects in natural populations, and adding epigenetics is likely to be a significant challenge. Again, I absolutely agree that the ecological relevance of patterns observed in the lab can be questioned, but from the perspective of understanding pure mechanism, they are likely to be our best way to make progress. I think it might be worth mentioning this caveat in section 5. It does not diminish from the overall message that studying natural populations is important. I also think it would be great if this section could be elaborated to discuss how studying particular ecological or environmental scenarios would contribute to knowledge about specific epigenetic processes. E.g., is there an example where a study in the wild yielded results that would not have been expected based on lab studies because of some natural process that was missing in the lab? Even if no example currently exists, perhaps a theoretical one could be described. This would help the reader understand why this natural context can add relevance that might be lacking in lab studies.

3. Regarding the point about filtering uninformative sites. I completely agree that the consequences of filtering is an important topic and has received too little attention, and I am not suggesting that this section be dropped. I am just uncertain about the statistical effects of what is being proposed in the current text. My main point was about the consequences of filtering out uninformative sites for the genome-wide threshold for significance. Again, I may not be

understanding things correctly, but I don't think it actually matters what proportion of the uninformative sites are 0% methylation vs. 100% methylation. The important thing is that they remain at 0% or 100% between the two comparison points (this is how I interpreted 'uninformative' in a test of differential methylation) and thus show no change (or difference). The threshold for significance will be determined by the distribution of methylation change (or difference) across sites. Leaving all of the sites which show no change in the dataset means you will have a distribution heavily weighted with a lot of zeros, which means that the informative sites that do show change now have a much easier time testing as outliers and therefore being considered as significant in a test of differential methylation. In an extreme where a large proportion of the sites in the dataset show no change/difference in methylation, even very minor differences in methylation could test as significant, likely leading to false positives. Am I misunderstanding this?

I also agree that, just like polygenic adaptation, widespread small effects are likely to be important for epigenetics. But given our current statistical methods, distinguishing these small effects from random noise is going to be quite difficult (the point above is related to this). Given the limitations of our current analytical frameworks for methylation data, for the immediate future we might make the most progress focusing on the meaningful effects that we can confidently distinguish from stochastic noise, and these are inevitably going to be (relatively) large. I am also hopeful that as methods develop we can move towards a better understanding poly(epi)genetics. And it is possible I am not aware of the latest methods for dealing with these issues. If there are methods analogous to the polygenic models of adaptation that have been developed for SNP data (e.g., by Gompert, Coop, and others) that can be used for methylation data then they should certainly be discussed here.

Review form: Reviewer 2

Recommendation

Accept as is

Scientific importance: Is the manuscript an original and important contribution to its field?

Good

General interest: Is the paper of sufficient general interest?

Good

Quality of the paper: Is the overall quality of the paper suitable?

Good

Is the length of the paper justified?

Yes

Should the paper be seen by a specialist statistical reviewer?

No

Do you have any concerns about statistical analyses in this paper? If so, please specify them explicitly in your report.

No

It is a condition of publication that authors make their supporting data, code and materials available - either as supplementary material or hosted in an external repository. Please rate, if applicable, the supporting data on the following criteria.

Is it accessible?

N/A

Is it clear?

N/A

Is it adequate?

N/A

Do you have any ethical concerns with this paper?

No

Comments to the Author

The author has addressed all my concerns. After correcting a number of typos and grammatical errors, the readability of the manuscript has improved.

Decision letter (RSPB-2021-1633.R0)

23-Aug-2021

Dear Professor Husby:

Your manuscript has now been peer reviewed and the reviews have been assessed by an Associate Editor. The reviewers' comments (not including confidential comments to the Editor) and the comments from the Associate Editor are included at the end of this email for your reference. As you will see, the reviewers and the Editors have raised some concerns with your manuscript and we would like to invite you to revise your manuscript to address them.

Research ethics:

Use of animals and field studies:

It is a condition of publication that you make available the data and research materials supporting the results in the article (<https://royalsociety.org/journals/authors/author-guidelines/#data>). Datasets should be deposited in an appropriate publicly available repository and details of the associated accession number, link or DOI to the datasets must be included in the Data Accessibility section of the article (<https://royalsociety.org/journals/ethics-policies/data-sharing-mining/>). Reference(s) to datasets should also be included in the reference list of the article with DOIs (where available).

Please submit a copy of your revised paper within three weeks. If we do not hear from you within this time your manuscript will be rejected. If you are unable to meet this deadline please let us know as soon as possible, as we may be able to grant a short extension.

Best wishes,
Professor Gary Carvalho

Associate Editor Board Member

Comments to Author:

Your revised manuscript has been reviewed by two reviewers who both agreed that the manuscript is improved. Reviewer 1 raised a few minor comments that, if addressed, have potential to further strengthen the manuscript. In particular, please address the second comment about elaborating on how studying epigenetics in natural populations has the potential to contribute to our understanding of epigenetic mechanisms important for evolution.

Reviewer(s)' Comments to Author:

Referee: 1

Comments to the Author(s).

I was previously reviewer 1 for this manuscript. I feel that the revisions have addressed a number of my previous concerns. I have a few additional comments:

1. While the addition of two figures helps address the lack of display elements in the manuscript, the two which have been added are relatively superficial: one which provides photos of organisms that have been the focus of epigenetic study and one providing a very basic schematic of interactions between the environment, genetics, epigenetics, and phenotype. These are nice, but I still feel it would be useful to add something more substantive (e.g., a table covering the specific strengths and weaknesses of different approaches mentioned, or their best areas of application, or a figure showing data from one of the key papers cited that can clearly convey one of the central points being made in the text).

2. The central critique in my initial review was that the review did not do an adequate job of articulating precisely how studying natural populations will help to understand the role of epigenetic mechanisms in generating phenotypic diversity. The revised section 5 now briefly addresses this issue by reiterating the point I made in my review, that "the ecological relevance of the environmental factors tested in the lab may be unclear, and the genetic backgrounds and epigenetic variation present in lab-raised organisms are often not representative of natural populations.", but, as I stated previously, this might not matter if the only goal is to obtain a mechanistic understanding of the interactions between the environment, genetic variation, epigenetic variation, and phenotype. We already have a difficult time disentangling genetic and environmental effects in natural populations, and adding epigenetics is likely to be a significant challenge. Again, I absolutely agree that the ecological relevance of patterns observed in the lab can be questioned, but from the perspective of understanding pure mechanism, they are likely to be our best way to make progress. I think it might be worth mentioning this caveat in section 5. It does not diminish from the overall message that studying natural populations is important. I also think it would be great if this section could be elaborated to discuss how studying particular ecological or environmental scenarios would contribute to knowledge about specific epigenetic processes. E.g., is there an example where a study in the wild yielded results that would not have been expected based on lab studies because of some natural process that was missing in the lab? Even if no example currently exists, perhaps a theoretical one could be described. This would help the reader understand why this natural context can add relevance that might be lacking in lab studies.

3. Regarding the point about filtering uninformative sites. I completely agree that the consequences of filtering is an important topic and has received too little attention, and I am not suggesting that this section be dropped. I am just uncertain about the statistical effects of what is being proposed in the current text. My main point was about the consequences of filtering out uninformative sites for the genome-wide threshold for significance. Again, I may not be understanding things correctly, but I don't think it actually matters what proportion of the uninformative sites are 0% methylation vs. 100% methylation. The important thing is that they remain at 0% or 100% between the two comparison points (this is how I interpreted 'uninformative' in a test of differential methylation) and thus show no change (or difference). The

threshold for significance will be determined by the distribution of methylation change (or difference) across sites. Leaving all of the sites which show no change in the dataset means you will have a distribution heavily weighted with a lot of zeros, which means that the informative sites that do show change now have a much easier time testing as outliers and therefore being considered as significant in a test of differential methylation. In an extreme where a large proportion of the sites in the dataset show no change/difference in methylation, even very minor differences in methylation could test as significant, likely leading to false positives. Am I misunderstanding this?

I also agree that, just like polygenic adaptation, widespread small effects are likely to be important for epigenetics. But given our current statistical methods, distinguishing these small effects from random noise is going to be quite difficult (the point above is related to this). Given the limitations of our current analytical frameworks for methylation data, for the immediate future we might make the most progress focusing on the meaningful effects that we can confidently distinguish from stochastic noise, and these are inevitably going to be (relatively) large. I am also hopeful that as methods develop we can move towards a better understanding poly(epi)genetics. And it is possible I am not aware of the latest methods for dealing with these issues. If there are methods analogous to the polygenic models of adaptation that have been developed for SNP data (e.g., by Gompert, Coop, and others) that can be used for methylation data then they should certainly be discussed here.

Referee: 2

Comments to the Author(s).

The author has addressed all my concerns. After correcting a number of typos and grammatical errors, the readability of the manuscript has improved.

Author's Response to Decision Letter for (RSPB-2021-1633.R0)

See Appendix B.

Decision letter (RSPB-2021-1633.R1)

27-Oct-2021

Dear Professor Husby:

Your manuscript has now been peer reviewed and the reviews have been assessed by an Associate Editor. The reviewers' comments (not including confidential comments to the Editor) and the comments from the Associate Editor are included at the end of this email for your reference. As you will see, the reviewers and the Editors have raised some concerns with your manuscript and we would like to invite you to revise your manuscript to address them.

Research ethics:

Use of animals and field studies:

It is a condition of publication that you make available the data and research materials supporting the results in the article (<https://royalsociety.org/journals/authors/author-guidelines/#data>). Datasets should be deposited in an appropriate publicly available repository and details of the associated accession number, link or DOI to the datasets must be included in the Data Accessibility section of the article (<https://royalsociety.org/journals/ethics-policies/data-sharing-mining/>). Reference(s) to datasets should also be included in the reference list of the article with DOIs (where available).

All supplementary materials accompanying an accepted article will be treated as in their final form. They will be published alongside the paper on the journal website and posted on the online

figshare repository. Files on figshare will be made available approximately one week before the accompanying article so that the supplementary material can be attributed a unique DOI. Please try to submit all supplementary material as a single file.

Please submit a copy of your revised paper within three weeks. If we do not hear from you within this time your manuscript will be rejected. If you are unable to meet this deadline please let us know as soon as possible, as we may be able to grant a short extension.

Best wishes,
Professor Gary Carvalho
Editor, Proceedings B
mailto:proceedingsb@royalsociety.org

Associate Editor
Board Member
Comments to Author:

I think the author has done a good job addressing reviewer comments. I agree with Reviewer 1 that a table on the specific strengths and weaknesses of different approaches for measuring epigenetic modifications would be very informative. Given the space constraints, I would strongly advocate for replacing Figure 1 (which is pretty but doesn't add much) with a table. The review is fine as-is, but I do think adding this table would substantially increase the usefulness of the review.

Also, please correct the many grammatical errors throughout the MS (see attachment).

Reviewer(s)' Comments to Author:

Author's Response to Decision Letter for (RSPB-2021-1633.R1)

See Appendix C.

Decision letter (RSPB-2021-1633.R2)

06-Jan-2022

Dear Professor Husby

I am pleased to inform you that your Review manuscript RSPB-2021-1633.R2 entitled "Wild epigenetics: insights from epigenetic studies on natural populations" has been accepted for publication in Proceedings B.

The referee(s) do not recommend any further changes. Therefore, please proof-read your manuscript carefully and upload your final files for publication. Because the schedule for publication is very tight, it is a condition of publication that you submit the revised version of your manuscript within 7 days. If you do not think you will be able to meet this date please let me know immediately.

To upload your manuscript, log into <http://mc.manuscriptcentral.com/prsb> and enter your Author Centre, where you will find your manuscript title listed under "Manuscripts with Decisions." Under "Actions," click on "Create a Revision." Your manuscript number has been appended to denote a revision.

You will be unable to make your revisions on the originally submitted version of the manuscript. Instead, upload a new version through your Author Centre.

1) A text file of the manuscript (doc, txt, rtf or tex), including the references, tables (including captions) and figure captions. Please remove any tracked changes from the text before submission. PDF files are not an accepted format for the "Main Document".

2) A separate electronic file of each figure (tiff, EPS or print-quality PDF preferred). The format should be produced directly from original creation package, or original software format. Please note that PowerPoint files are not accepted.

3) Electronic supplementary material: this should be contained in a separate file from the main text and the file name should contain the author's name and journal name, e.g. `authorname_procb_ESM_figures.pdf`

All supplementary materials accompanying an accepted article will be treated as in their final form. They will be published alongside the paper on the journal website and posted on the online figshare repository. Files on figshare will be made available approximately one week before the accompanying article so that the supplementary material can be attributed a unique DOI. Please see: <https://royalsociety.org/journals/authors/author-guidelines/>

4) Data-Sharing and data citation

It is a condition of publication that data supporting your paper are made available. Data should be made available either in the electronic supplementary material or through an appropriate repository. Details of how to access data should be included in your paper. Please see <https://royalsociety.org/journals/ethics-policies/data-sharing-mining/> for more details.

<http://datadryad.org/submit?journalID=RSPB&manu=RSPB-2021-1633.R2> which will take you to your unique entry in the Dryad repository.

Once again, thank you for submitting your manuscript to Proceedings B and I look forward to receiving your final version. If you have any questions at all, please do not hesitate to get in touch.

Sincerely,
Professor Gary Carvalho

Editor, Proceedings B
mailto:proceedingsb@royalsociety.org

Associate Editor
Comments to Author:

The author has done a great job addressing reviewer comments. I think the only thing I would add is it may be nice to add citations to the papers introducing the different methods to Table 1. Thank you for a very nice contribution to our special issue!

Decision letter (RSPB-2021-1633.R3)

12-Jan-2022

Dear Professor Husby

I am pleased to inform you that your manuscript entitled "Wild epigenetics: insights from epigenetic studies on natural populations" has been accepted for publication in Proceedings B.

Data Accessibility section

Open Access

Paper charges

Sincerely,
Proceedings B
<mailto:proceedingsb@royalsociety.org>

Appendix A

Dear Editors,

Please find resubmission for manuscript RSPB-2020-2838 entitled: “Wild epigenetics: insights from epigenetic studies on natural populations” for potential contribution to the special issue in “Wild Quantitative Genetics” attached.

I am very grateful to the two reviewers and the editor for helpful comments and thoughts on the manuscript and have followed all their advice. In particular reviewer 1 made many helpful suggestions on how to improve the delivery and also the writing of the manuscript and I think it is much improved as a result.

More specifically I have:

- 1) Expanded and revised the text to make it more clear why studies on natural populations are necessary (section 5)
- 2) Elaborated how a quantitative genetic approach on natural populations are important for studying epigenetic patterns (section 5)
- 3) Included two new figures
- 4) Revised the text throughout and included some of the suggested references as well as some new recent studies that have been published since the original submission.

Detailed responses to editor and reviewers comments can be found below with my response highlighted in red font.

Please do not hesitate to contact me if you have any questions and I look forward to hearing from you.

Best wishes,

Arild

Response to editor and reviewers

Associate Editor

Board Member: 1

Comments to Author:

Dear Dr. Husby

Thank you for your submission to our special issue on Wild Quantitative Genetics in Proc R Soc B. Your manuscript has now been reviewed by two experts in the field. Both reviewers were positive about some aspects of the manuscript, especially its discussion of the strengths and weaknesses of different methods, which they thought would be even more strong with a

figure or table. However, one of the reviewers had a number of critiques that should be addressed. In particular, I agree that further elaborating the importance of studying epigenetics in wild populations and describing how this could be accomplished with a quantitative genetics approach would improve the manuscript.

Thank you for the possibility to resubmit this work – I have now elaborated on the importance of studying epigenetics in wild populations and how a quantitative genetic approach is useful for this as recommended by the editor as well as reviewer 1. This can be found in sections 5 and 6 as well as brief mentioning also in other places of the manuscript.

Reviewer 1

I feel that the strongest element of the paper is the discussion on the strengths and weaknesses of different methodological approaches for conducting epigenetics with natural populations. This is an area that has not been covered in as much detail in previous work and will be very useful for molecular ecologists that are new to epigenetics and need to make practical decisions about study design and methodology. Methods in this field are also moving very quickly such that any discussion of sequencing approaches in papers from even just a few years ago begins to lose relevance.

Below I provide some comments that I hope will be useful.

I am very grateful to the reviewer for this very thoughtful and helpful review that has greatly increased the quality of the manuscript!

General: A central challenge that is highlighted in the review is the need to identify the role of epigenetic mechanisms operating independently of genetic variation. However, although it is stated repeatedly that studying natural populations will be important for achieving this goal, I don't feel the paper really articulates why. If anything, one would expect that this precise challenge would best be addressed by careful studies under controlled lab conditions, which could best disentangle environmental, genetic, and epigenetic effects on phenotypic variation. Of course, the ecological relevance of the environmental factors tested in the lab may be unclear, and the genetic backgrounds and epigenetic variation present in lab-raised organisms are often not representative of natural populations. But this might not matter if the only goal is to obtain a mechanistic understanding of the interactions between the environment, genetic variation, and epigenetic variation. This is not to say that studies in natural populations are not needed – they clearly are. But I don't think this paper makes a convincing case why. What are the specific situations in which natural populations provide novel insights into epigenetic mechanisms that we cannot attain through lab studies? It seems important for a review on this topic to clearly articulate this.

This is an important point and I have now revised the manuscript in several places to try to make a more convincing case on why this is important. In particular I have developed section 5 and 6 more.

General: I'm afraid the paper is not very well-written, with unclear wording in many sections, repetitive sentences, and numerous grammar mistakes and typos that sometimes impede

interpretation of the text. I have done my best to flag these. A revised manuscript would benefit from a thorough re-write to improve the clarity and structure of the paper.

I appreciate the helpful comments for revising the text and the many typos the reviewer has helped to weed out – I hope readability is now improved.

General: It might be useful to have a table summarizing the characteristics of the various methods described in the paper. There are no display items of any kind in the paper, which is unusual. For instance, it might also be nice to include a figure showing a key result from one of the empirical papers that is cited to help illustrate the point being made. Or some kind of conceptual diagram to help demonstrate how genetic and epigenetic mechanisms could independently contribute to phenotypic variation, etc.

It was an error on my side not including figures, something also pointed out by reviewer 2. I have now included two figures, one highlighting the study organisms that are setting the tone for epigenetic studies in the wild and one figure to illustrate how a quantitative genetic approach can be used to provide further insights into epigenetic mechanisms in natural populations.

Section 1. “The emerging role of epigenetic mechanisms in generating phenotypic diversity”: I don’t think this section actually tells us anything about the role of epigenetic mechanisms in generating phenotypic diversity. It describes how work in this area is increasing and outlines some methodological advances but doesn’t explain what they have revealed. It also seems like it would be useful to provide some basic definitions of epigenetic terms/mechanisms in this section and explain how they can generate phenotypic diversity. E.g., how do histone modifications or small RNAs change phenotype? This will be helpful for newcomers to epigenetics.

It was my intention to have a more historical account and guide the readers to the developments that have led to the current state of the field today. I have revised the heading so that it is not misleading as to its content.

L19: Use of “also” is unclear. Presumably this is meant to convey that epigenetic mechanisms are now increasingly studied in natural populations as well as lab-reared organisms, but coming without context in the first sentence of the abstract this is not clear.

Revised

L24: “much of this is” -> “many of these studies are”

Done

L25: “contribution” -> “contributions”

Done

L27: “resolve” -> “resolving”

Done

L28: Incorrect grammar.

Revised

L37: “#REF”. Please add the citations.

Revised

L39: “at” -> “with”

Done

L41: “mid 2000” -> “the mid 2000s”

Done

L44: Here and in several other places in the paper the use of “also” is unclear. It would be good to be more precise about what is being compared with. Presumably model species is meant here, but the wording is vague (e.g., this could also be drawing a contrast with scales lower than genome-wide).

That is a good point, I have now revised this here and in other places.

L50: “natural population” -> “natural populations”

Done

L51: “techniques” -> “technique”

Done

L51: methylation spelled wrong.

Done

L53: “advancement” -> “advancements”

Done

L64: Typo, Incorrect grammar.

Done

L71: Perhaps define ‘peloric’.

Done

L73: The publication of a single paper doesn’t necessarily put something “firmly on the map”. How did it change opinions? Was it well-cited? Did many other papers on the topic soon follow?

Revised, it is highly cited (~1200 times).

L79: “seem to contribute”. But to what extent? Here and in other places it would be useful to provide a bit more nuance and insight about the relative roles of different mechanisms. Few people would dispute that epigenetic mechanisms can contribute in some way to phenotypic variation. The more interesting question (and one that would hopefully be addressed in a review on this specific topic) is whether it is a meaningful contribution. Or if the relative roles of epigenetic vs. genetic variation shift under particular ecological or evolutionary circumstances.

Revised. I want to highlight that both can act in concert here as I return to this point several other places in the review

L83: “sate” -> “state”.

Done

L92: “plants” -> “plant”

Done

L109: “here studies on natural populations can play a key role”. Again, it is unclear why natural populations are needed for addressing issues this specific issue. In many ways I would expect a controlled lab study to be best suited for isolating the impact of transgenerational epigenetic effects on phenotypic variation.

I address this in section 5

L121: The bisulphite conversion is not specific to reduced representation approaches. This could be moved to more general section of text.

I mention this for both the reduced representation and whole genome approach with Illumina to make the point that long read seq techniques don't require this so I think its important to bring up in both places

L141: “require” -> “requires”

Done

L145: Define “epiGBS”. There is inconsistency about which terms/acronyms are defined and which are not (e.g., RRBS is, but epiGBS, ddRAD, and ATAC-seq are not).

Now defined

L150: It might be good to explain why interest in CpG methylation would be associated with study system.

Revised

L156: For consistency, state whether epiRAD requires a reference genome.

Added

L158: It is not ideal to have a single sentence paragraphs like this (here and elsewhere in the paper). Perhaps try to merge them with other text.

Revised

L167: “offer” -> “offers”

Done

L172: The context for this result is unclear. Is this in a single individual? What is the generality?

The cited study provide one example of tissue specific differences in non-CpG context, most studies look at CpG and taking a genome wide approach is one benefit which allow non CpG information to be gathered.

L198: “is” -> “are”. Might be nice to list what these projects are.

Removed sentence

L206: Explain why less input material is required.

I do not think it is necessary to go into the details of the protocols here – interested readers can find that from the references

L208: “control” -> “controls”

Done

L211: “even have consequences for population demography”. This is unclear. In addition to what?

Now revised this sentence

L213: This same example was already given and described earlier in the paper.

Yes, but since I go into more detail here I have kept it

L216: Explain what RNAi is.

Done

L225: This is a bit repetitive with what has come earlier.

I agree and have now removed this paragraph.

L226: A single individual could potentially be used, but the generality of the inferences would be limited.

Yes, revised sentence

L234: “reduced” -> “reduced representation”.

Done

L241: Incorrect grammar.

Revised

L244: These two citations don’t necessarily indicate those fragment sizes are “commonly” used in birds or mammals – only a single species is used in each of the papers.

Good point, I have revised the sentence

L255: Incorrect grammar.

Revised

L258: “receptive” -> “repetitive”

Done

L260: Incorrect grammar.

Revised

L265: Other enzymes could also be used.

Indeed, revised

L321: “typically” and “most commonly” are redundant.

Done

L334: “play” -> “plays”

Done

L344: “exists” -> “exist”

Done

L346: “consist” -> “consists”

Done

L356: Incorrect grammar.

Revised

L365: Incorrect grammar.

Revised

L367: “show” -> “shows”

Done

L370: “(always unmethylated)”. My understanding is that hypomethylation and hypermethylation of DNA are relative terms and denote less or more methylation when compared to some DNA standard, so use of hypomethylation would not necessarily indicate that a site is always unmethylated (at least at a population level).

Both definitions are used – I have revised sentence to make this clear

L375: “p-value” -> “p-values”

Done

L379: I might not be thinking about this clearly, but the logic here seems backwards. Wouldn't leaving the uninformative sites in the dataset be more likely to result in test statistic inflation? Excluding these sites prior to statistical testing seems like a sensible approach to avoid false positives for significant hyper/hypo-methylation. If they are left in the dataset, they will indeed reduce the proportion of sites that show a change in methylation since the denominator will now include a large number of sites showing no change. But, these sites will also reduce the genome-wide threshold for significance to make it easier for any individual site that does show a change in methylation to test as being significantly differentially methylated. I think most researchers are more interested in confidently identifying large and meaningful changes in methylation, as opposed to knowing what proportion of sites show a change vs. no change (since the magnitude of many of the changes will be small and therefore will not have meaningful impacts). As an analogy, one would not want to identify statistically significant *Fst* outliers using a dataset that contains uniformly invariant sites.

Why would you get false positives for significant hyper/hypomethylation in sites that do not change more than in other parts of the genome? If we assume 10,000 sites are tested and 5,000 of them are consistently hypo or hypermethylated (so either no or 100% methylation) then removing these 5,000 sites will impact the genome wide methylation estimate and it will only be unbiased if 2,500 have 0% methylation and 2,500 have 100%. Any change from this leads to over or underestimation of methylation differences.

I am also not so sure that most are interested in identifying “large and meaningful” changes in methylation – we now know polygenic adaptation is widespread and I would not be surprised if this extended to methylation studies and thus many and small changes might be equally important.

In my view the consequences of the filtering step studies use has received too little attention and therefore I think it is useful to point this out.

L390: Measuring selection does not need to be done in the natural environment of the organism. This is required for measuring ecologically-relevant natural selection, but not ‘selection’.

Good point – now revised

L391: And phenotypic data as well, presumably.
yes, revised

L393: This example directly contradicts the opening point made for this section. The A. thaliana experiment was done under controlled conditions, not natural conditions.

Yes, although in a simulated fragmented landscape so not the typical controlled conditions. I also emphasized that on line 405 but have now further made that clear. I included this example to demonstrate that selection can act on epigenetic diversity as very few studies have estimated selection on epigenetic diversity (many infer it but few have demonstrated it).

L394: Incorrect grammar.

Revised

L410: Why do you find them the most interesting?

For example because selection cant act on them if they don't have a phenotypic effect but I have now revised the sentence.

L426: “induces” -> “induced”

Done

L429: Citations?

This is a statement of what epigenetic studies on natural populations can provide insights into...

L441: “for” -> “of”

Done

L443: It would be nice to explain how this is accomplished.

This is explained on 435-439

L448: Wording could be improved.

Revised

L452: Incorrect grammar/wording.

Revised

L454: It would be useful to explain this a bit more for readers who are not familiar with the resetting of epigenetic marks. E.g., what proportion is typically re-set? Does this vary across groups? Does it vary based on environmental conditions?

I have now removed that statement as I revised the paragraph

L459: This is a bit vacuous. It could potentially contribute to local adaptation or it could not. It would depend on a number of factors that aren't really explained.

Vacuous indeed (and new word learned), now revised

L468: Incorrect grammar.

Revised

L472: Remove "to".

Done

L475: "mechanism" -> "mechanisms"

Done

References: A mix of formatting styles are used and there are spelling mistakes throughout.

Revised

Below are some relevant studies on epigenetics in natural populations or under ecologically relevant conditions that are not cited. These are just papers that I remember and is not an exhaustive list. But given this is a formal review paper, it would be good to cover as much of the relevant literature as possible given this is a fairly new field and there are still not very many data papers on the topic.

Yes, I do not claim to have included all relevant papers, even in this relatively young field this is not feasible in the allowed space. I have however included some of the references below as well as a few new ones that have come out in the time since submission that I find particularly important. An exhaustive list of all published ecological epigenetic studies is not the goal however.

McNew SM, Beck D, Sadler-Riggelman I, Knutie SA, Koop JAH, Clayton DH, Skinner MK. 2017. Epigenetic variation between urban and rural populations of Darwin's finches. BMC Evol Biol. 17:183.

Baerwald MR, Meek MH, Stephens MR, Nagarajan RP, Goodbla AM, Tomalty KM, Thorgaard GH, May B, Nichols KM. 2016. Migration-related phenotypic divergence is associated with epigenetic modifications in rainbow trout. Mol Ecol. 25(8):1785–1800.

Le Luyer J, Laporte M, Beacham TD, Kaukinen KH, Withler RE, Leong JS, Rondeau EB, Koop BF, Bernatchez L. 2017. Parallel epigenetic modifications induced by hatchery rearing in a Pacific salmon. Proc Natl Acad Sci U S A. 114(49):12964–12969.

Hu J, Askary AM, Thurman TJ, Spiller D, Palmer TM, Pringle RM, Barrett RDH. 2019. Epigenetic signatures of colonizing new environments in Anolis lizards. Mol Biol Evol 36(10): 2165-2170.

- Huang X, Li S, Ni P, Gao Y, Jiang B, Zhou Z, Zhan A. 2017. Rapid response to changing environments during biological invasions: DNA methylation perspectives. *Mol Ecol.* 26(23):6621–6633.
- Uren Webster TM, Rodriguez-Barreto D, Martin SAM, van Oosterhout C, Wengel P, Cable J, Hamilton A, Garcia de Leaniz C, Consuegra S. 2018. Contrasting effects of acute and chronic stress on the transcriptome, epigenome, and immune response of Atlantic salmon. *Epigenetics* 13(12):1191–1207.
- Liebl, A.L., Schrey, A.W., Richards, C.L. & Martin, L.B. 2013. Patterns of DNA methylation throughout a range expansion of an introduced songbird. *Integr. Comp. Biol.* 53: 351–358.
- Liu, S., Sun, K., Jiang, T., Ho, J.P., Liu, B. & Feng, J. 2012. Natural epigenetic variation in the female great roundleaf bat *Hipposideros armiger* populations. *Mol. Genet. Genomics* 287: 643–650.
- Blouin, M.S., Thuilier, V., Cooper, B., Amarasinghe, V., Cluzel, L., Hitoshi, A. et al. 2010. No evidence for large differences in genomic methylation between wild and hatchery steelhead *Oncorhynchus mykiss*. *Can. J. Fish Aquat. Sci.* 67: 217–224.
- Dimond, J.L. & Roberts, S.B. 2016. Germline DNA methylation in reef corals: patterns and potential roles in response to environmental change. *Mol. Ecol.* 25: 1895–1904.
- Dubin, M.J., Zhang, P., Meng, D., Remigereau, M.S., Osborne, E.J., Paolo, C.F. et al. 2015. DNA methylation in *Arabidopsis* has a genetic basis and shows evidence of local adaptation. *eLife* 4: e05255.
- Foust, C.M., Preite, V., Schrey, A.W., Alvarez, M., Robertson, M.H., Verhoeven, K.J.F. et al. 2016. Genetic and epigenetic differences associated with environmental gradients in replicate populations of two salt marsh perennials. *Mol. Ecol.* 25: 1639–1652.
- Gao, L., Geng, Y., Li, B., Chen, J. & Yang, J. 2010. Genome-wide DNA methylation alterations of *Alternanthera philoxeroides* in natural and manipulated habitats: implications for epigenetic regulation of rapid responses to environmental fluctuation and phenotypic variation. *Plant, Cell Environ.* 33: 1820–1827.
- Gugger, P., Fitz-Gibbon, S., Pellegrini, M. & Sork, V.L. 2016. Species-wide patterns of DNA methylation variation in *Quercus lobata* and its association with climate gradients. *Mol. Ecol.* 25: 1665–1680.
- Keller, T.E., Lasky, J.R. & Yi, S.V. 2016. The multivariate association between genome-wide DNA methylation and climate across the range of *Arabidopsis thaliana*. *Mol. Ecol.* 25: 1823–1837.
- Lea, A.J., Altmann, J., Alberts, S.C. & Tung, J. 2016. Resource base influences genome-wide DNA methylation levels in wild baboons *Papio cynocephalus*. *Mol. Ecol.* 25: 1681–1696
- Verhoeven, K.J.F., Jansen, J.J., van Dijk, P.J. & Biere, A. 2010. Stress-induced DNA methylation changes and their heritability in asexual dandelions. *New Phytol.* 185: 1108–1118.

Reviewer 2

See below my detailed comments.

1) I strongly believe a figure or diagram is needed to recapitulate the review's major contents. For example, list the pros and cons of the different approaches for measuring DNA methylation in natural populations would be helpful.

This is a good suggestion, I have now added two figures that illustrate different aspects of DNA methylation studies on wild populations.

2) I'd appreciate it if the author can generalize the epigenetic measurement section (from line 112) to include not only animals but plants.

Good point, I have now revised this part.

3) Some recent publications highly relevant to the current review should consider being cited.

Shahryary, Y., Symeonidi, A., Hazarika, R.R. et al. AlphaBeta: computational inference of epimutation rates and spectra from high-throughput DNA methylation data in plants. *Genome Biol* 21, 260 (2020). <https://doi.org/10.1186/s13059-020-02161-6>

Xu, G., Lyu, J., Li, Q. et al. Evolutionary and functional genomics of DNA methylation in maize domestication and improvement. *Nat Commun* 11, 5539 (2020). <https://doi.org/10.1038/s41467-020-19333-4>

Thank you for the suggestions, I have included several new citations in the revised version.

Appendix B

Uppsala
6th October 2021

Dear Editors,

Please find resubmission for manuscript RSPB-2020-2838 entitled: “Wild epigenetics: insights from epigenetic studies on natural populations” for potential contribution to the special issue in “Wild Quantitative Genetics” attached.

I am very grateful to the two reviewers and the editor for helpful comments and thoughts on the manuscript and have followed all their advice. In particular reviewer 1 made many helpful suggestions on how to improve the delivery and also the writing of the manuscript and I think it is much improved as a result.

Detailed responses to editor and reviewers comments can be found below with my response highlighted in red font.

Please do not hesitate to contact me if you have any questions and I look forward to hearing from you.

Best wishes,

Arild

Response to editor and reviewers (in bold)

Your revised manuscript has been reviewed by two reviewers who both agreed that the manuscript is improved. Reviewer 1 raised a few minor comments that, if addressed, have potential to further strengthen the manuscript. In particular, please address the second comment about elaborating on how studying epigenetics in natural populations has the potential to contribute to our understanding of epigenetic mechanisms important for evolution.

Thank you for the opportunity to resubmit, I have now included more discussion of how studying epigenetics in natural population can contribute to our understanding of the role that epigenetics can play in evolution.

Reviewer(s)' Comments to Author:

Referee: 1

Comments to the Author(s).

I was previously reviewer 1 for this manuscript. I feel that the revisions have addressed a number of my previous concerns. I have a few additional comments:

1. While the addition of two figures helps address the lack of display elements in the manuscript, the two which have been added are relatively superficial: one which provides photos of organisms that have been the focus of epigenetic study and one providing a very basic schematic of interactions between the environment, genetics,

epigenetics, and phenotype. These are nice, but I still feel it would be useful to add something more substantive (e.g., a table covering the specific strengths and weaknesses of different approaches mentioned, or their best areas of application, or a figure showing data from one of the key papers cited that can clearly convey one of the central points being made in the text).

There is not really much space to add additional display items since I am already at the estimated ten page limit in PRSB. I have there chosen not to include additional tables and display items.

2. The central critique in my initial review was that the review did not do an adequate job of articulating precisely how studying natural populations will help to understand the role of epigenetic mechanisms in generating phenotypic diversity. The revised section 5 now briefly addresses this issue by reiterating the point I made in my review, that “the ecological relevance of the environmental factors tested in the lab may be unclear, and the genetic backgrounds and epigenetic variation present in lab-raised organisms are often not representative of natural populations.”, but, as I stated previously, this might not matter if the only goal is to obtain a mechanistic understanding of the interactions between the environment, genetic variation, epigenetic variation, and phenotype. We already have a difficult time disentangling genetic and environmental effects in natural populations, and adding epigenetics is likely to be a significant challenge. Again, I absolutely agree that the ecological relevance of patterns observed in the lab can be questioned, but from the perspective of understanding pure mechanism, they are likely to be our best way to make progress. I think it might be worth mentioning this caveat in section 5. It does not diminish from the overall message that studying natural populations is important. I also think it would be great if this section could be elaborated to discuss how studying particular ecological or environmental scenarios would contribute to knowledge about specific epigenetic processes. E.g., is there an example where a study in the wild yielded results that would not have been expected based on lab studies because of some natural process that was missing in the lab? Even if no example currently exists, perhaps a theoretical one could be described. This would help the reader understand why this natural context can add relevance that might be lacking in lab studies.

Yes, we both agree on this point and I stress the difficulty in separating epigenetic and genetic effects several places in the ms. I fully agree that there is a strong value also in lab based research (In my own research I collect individuals from the wild into the lab for functional work exactly for this reason) and I have now also added an explicit mentioning of this in part 5 (380-388).

3. Regarding the point about filtering uninformative sites. I completely agree that the consequences of filtering is an important topic and has received too little attention, and I am not suggesting that this section be dropped. I am just uncertain about the statistical effects of what is being proposed in the current text. My main point was about the consequences of filtering out uninformative sites for the genome-wide threshold for significance. Again, I may not be understanding things correctly, but I don't think it actually matters what proportion of the uninformative sites are 0% methylation vs. 100% methylation. The important thing is that they remain at 0% or 100% between the two comparison points (this is how I interpreted 'uninformative' in a test of differential methylation) and thus show no change (or difference). The threshold for significance will be determined by the distribution of methylation change (or difference) across sites. Leaving all of the sites which show no change in the dataset means you will have a distribution heavily weighted with a lot of zeros, which means that the informative sites that do show change now have a much easier time testing as outliers and therefore being considered as significant in a test of differential methylation. In an extreme where a large proportion of the sites in the dataset show no change/difference in methylation, even very minor differences in methylation could test as significant, likely leading to false positives. Am I misunderstanding this?

Its not quite correct what is stated above. The null model in the statistical tests are that p-values should be uniformly distributed when there are no statistically significant effects (ie when all null models are true). The reason for this is that when we define a certain p-value threshold to be, say, 5% we then reject that when the observed p-value is less than 5%. The point of using a particular distribution (t, F, chisquare etc) is to transform from the test statistics to a uniform p-value distribution. If the null hypothesis is false it will be weighted towards zero (which is were your 'significant' effects are). Now, if you have a lot of sites that do not change they will show up as enrichment in the far end of the p-value histogram distribution below (circled in red) and that means one violates the assumption of uniform p-value distribution. So that is the problem and why people then remove sites that are less than, say 15% differentially methylated, because if you did not you would have a very skewed p-value distribution when the statistical tests assume near uniformity. My point is that, as far as I am aware, the consequence of removing these sites for drawing statistical inference about which fraction of sites deviate from uniformity are poorly explored.

I also agree that, just like polygenic adaptation, widespread small effects are likely to be important for epigenetics. But given our current statistical methods, distinguishing these small effects from random noise is going to be quite difficult (the point above is related to this). Given the limitations of our current analytical frameworks for methylation data, for the immediate future we might make the most progress focusing on the meaningful effects that we can confidently distinguish from stochastic noise, and these are inevitably going to be (relatively) large. I am also hopeful that as methods develop we can move towards a better understanding poly(epi)genetics. And it is possible I am not aware of the latest methods for dealing with these issues. If there are methods analogous to the polygenic models of adaptation that have been developed for SNP data (e.g., by Gompert, Coop, and others) that can be used for methylation data then they should certainly be discussed here.

To my knowledge there are no such models

Referee: 2

Comments to the Author(s).

The author has addressed all my concerns. After correcting a number of typos and grammatical errors, the readability of the manuscript has improved.

Thank you

Appendix C

UPPSALA
UNIVERSITET

Uppsala
8th December 2021

Dear Editors,

Please find resubmission for manuscript RSPB-2020-2838 entitled: “Wild epigenetics: insights from epigenetic studies on natural populations” for potential contribution to the special issue in “Wild Quantitative Genetics” attached.

I am very grateful to reviewers and the editor for the helpful comments and have now replaced Figure 1 with a table summarizing the advantages and disadvantages of different methods as suggested. I have also revised the text and implemented the changes suggested by the editor.

I apologize for the delay in returning the submission and thank you for your patience.

Please do not hesitate to contact me if you have any questions and I look forward to hearing from you.

Best wishes,

Arild